# Minimizing the electrosorption of water from humid ionic liquids on electrodes

Sheng Bi [1], Runxi Wang[1], Shuai Liu [2], Jiawei Yan [2], Bingwei Mao[2], Alexei A. Kornyshev [3] & Guang Feng [1]

In supercapacitors based on ionic liquid electrolytes, small amounts of absorbed water could potentially reduce the electrochemical window of electrolytes and cause performance degradation. The same would take place if ionic liquids are used as solvents for electro-catalysis involving the dissolved molecular species. In this work, we carry out molecular dynamics simulations, with gold and carbon electrodes in typical ionic liquids, hydrophobic and hydrophilic, to study electrosorption of water. We investigate the effects of hydro-phobicity/hydrophilicity of ionic liquids and electrodes on interfacial distribution of ions and electrosorbed water. Results reveal that using hydrophilic ionic liquids would help to keep water molecules away from the negatively charged electrodes, even at large electrode polarizations. This conclusion is supported by electrochemical cyclic voltammetry mea-surements on gold and carbon electrodes in contact with humid ionic liquids. Thereby, our findings suggest potential mechanisms for protection of electrodes from water electrosorption.

[1] State Key Laboratory of Coal Combustion, School of Energy and Power Engineering, Huazhong University of Science and Technology (HUST), 430074 Wuhan, China. [2] State Key Laboratory for Physical Chemistry of Solid Surfaces, and Department of Chemistry, College of Chemistry and Chemical Engineering, Xiamen University, 361005 Xiamen, China. [3] Department of Chemistry, Faculty of Natural Sciences, Imperial College London, London SW7 2AZ, UK. Correspondence and requests for materials should be addressed to G.F. (email: gfeng@hust.edu.cn) or to J.Y. (email: jwyan@xmu.edu.cn) or to A.A.K. (email: a.kornyshev@imperial.ac.uk)

Interest in using renewable energy resources, such as solar, wind, and tide, to alleviate global climate change and achieve sustainable development is growing in recent decades. Due to intermittency for many renewable energy sources, energy storage devices are in demand and would play a key role in increasing the portion of these intermittent renewables in electricity production[1,2]. Among energy storage solutions, electrical double layer (EDL) capacitors, also known as supercapacitors, have drawn much attention due to their high power density and ultra-long cycle life[3–5] since they store the charges through ion adsorption at electrified electrode surfaces. As the stored energy is controlled by EDLs, the electrolyte–electrode interfaces are essential and directly affect the performance of supercapacitors[6–8].

Among electrolytes used in supercapacitors, room temperature ionic liquids (RTILs) are an emerging class of ionic materials with exceptional characteristics, including low volatility, high thermal stability, and wide electrochemical window[9–11]. In particular, their wide electrochemical window helps to increase the operational voltage, which directly improves the energy density of supercapacitors. However, exploiting RTILs in a practical system is commonly more intricate than it seems. One of such issues is due to the hygroscopic nature of RTILs, which could make water originate from the synthesis or absorbed from humid air during fabrication or utilization. Many RTILs are water miscible and conventionally labeled as hydrophilic; for instance, the "popular" 1-butyl-3-methylimidazolium tetrafluoroborate ([BMIM][BF$_4$])[12,13]. However, even some RTILs that are more hydrophobic than others, such as water-immiscible RTIL 1-butyl-1-methylpyrrolidinium bis(trifluoromethylsulfonyl)imide ([pyr14][TFSI]), are still hygroscopic; they absorb water up to their water concentration saturation level[12–14]. Thus, one of the key advantages of RTILs for use as electrolytes in supercapacitors and electrocatalysis, their wide electrochemical window, could potentially be compromised due to the presence of water[15]. It would also be vital to many other applications with RTIL-based EDLs, such as lithium batteries[16], solar cells[17], electrowetting[18], RTIL-gated field-effect transistors[19], and electrochemical sensors[20].

Recently, however, reports on water-in-salt electrolytes have drawn much attention in the battery community[21–24]. The nature of the water effect has yet to be properly understood; it may be related with freezing out orientational degrees of freedom of water molecules needed for acquiring best conformations for reactions or due to hampering the formation of reaction intermediates, both of which could be more pronounced in intercalation confinement. On the other hand, at flat electrodes, data, including some in this paper, signify that water starts to react at electrodes, dramatically reducing the electrochemical window of the mixture[25,26].

Extensive theoretical and experimental studies have investigated the structure of EDLs at charged interfaces and in nanoconfinement, including configurations typical for supercapacitors with dry RTILs as electrolytes[27–34]. Recently, several studies have been devoted to investigating the effect of small amounts of water absorbed in RTILs. Based on molecular dynamic (MD) simulations, our previous work predicted that water in the hydrophobic RTIL 1-butyl-3-methylimidazolium hexafluorophosphate ([BMIM][PF$_6$]) can be electrically attracted by polarized carbon electrodes. Even if the water content for ionic liquids is quite small (i.e., ppm), it appears that water molecules can still accumulate on electrified interfaces when electrodes are polarized positively and negatively, which may potentially result in water electrolysis[35]. Similar behavior has been found also using MD simulations in nanoconfinement[36]. Experimentally, Motobayashi and Osawa[37] used surface-enhanced infrared absorption spectroscopy to study the humid 1-butyl-3-methylimidazolium bis(trifluoromethylsulfonyl)imide ([BMIM][TFSI])/gold interface,

confirming that water molecules accumulate at the gold surface and showing that the water condensation is dependent on potential (however, their method did not have atomic resolution and could not show in which layer of ionic liquid that water accumulates most)[37]. By using an atomic force microscopy (AFM) technique, Valtiner and coworkers[38] have investigated the influence of water on ion layering of [EMIM][TFSI] at charged gold surfaces. They did not observe evidence of electrosorption of water on ions layered at the ionophilic gold electrodes. However, AFM force curves showed that the incorporated water molecules weaken the stability of layered structures[39]. Despite these modeling and experimental studies, some critical questions, regarding the electrosorption of water on electrodes, remained unanswered, such as (1) effects of the nature of electrode materials, from gold to carbon, on water electrosorption, (2) differences between hydrophobic and hydrophilic RTILs, and (3) how to prevent water adsorption on electrode surfaces to avoid electrochemical reaction. In an attempt to answer these questions, we performed a series of MD simulations to investigate the effects of different electrodes and RTILs on interfacial ions and water distribution, as well as pertinent cyclic voltammetry (CV) measurements.

The MD cells considered in this work consist of a slab of humid RTILs enclosed between two oppositely charged electrodes (Supplementary Note 1, Supplementary Figure 1). Two kinds of electrodes (gold and carbon) and two major kinds of RTILs ([pyr14][TFSI] and [BMIM][BF$_4$]) are adopted, as they have been widely studied in previous experiments and simulations[6,25,28,40]. In particular, gold is known to be more ionophilic than carbon[41,42], while [pyr14][TFSI] is hydrophobic and [BMIM][BF$_4$] is hydrophilic[12]. The hydrophobicity and hydrophilicity as well as hygroscopicity of RTILs were examined by both MD simulation and experiment of water–RTIL mixture (Supplementary Note 2, Supplementary Figures 2 and 3), showing that the hydrophobic RTIL [pyr14][TFSI] is immiscible with water, while the hydrophilic RTIL [BMIM][BF$_4$] is water miscible (Supplementary Note 2, Supplementary Movies 1 and 2). By considering RTILs with humidity at gold and carbon electrodes (details in Supplementary Table 1), we show that the hydrophobicity/hydrophilicity of RTILs could significantly influence the electrosorption of incorporated water. We verify this fact by the corresponding CV measurements. On the other hand, the nature of the electrode appears to play only a minor role. Molecular insights into EDL structures and the mechanism of how water molecules behave near the electrified interfaces are clarified and discussed.

## Results

**Identification of key factors for water electrosorption.** We begin our work by investigating the effect of electrode material on EDL structure, as gold and carbon electrodes interact differently with ions of the studied RTILs. The EDL potential is defined as the potential across the EDL relative to its value at zero charge of the electrode: $\Phi_{EDL} = \Phi_{electrode} - \Phi_{bulk} - (\Phi_{electrode} - \Phi_{bulk})|_{pzc}$, where potential of zero charge (pzc) in MD simulation is well evaluated from the Poisson potential drop across the EDL, when the electrode surface charge is zero. Fig. 1 shows the number density profiles of ions (pyr14$^+$ and TFSI$^-$) as a function of distance from the electrode with varying the EDL potential. Qualitatively, both gold and carbon electrodes lead to strong ion layering, and as expected, counter-ions accumulate near the surfaces as the EDL potential increases. By comparing Fig. 1a with Fig. 1c for cations and Fig. 1b with Fig. 1d for anions, we can see that both cations and anions have much stronger adsorption on gold electrode than on carbon electrode; in other words, gold is, as expected, more ionophilic than carbon[41,42]. This is especially pronounced for cations: although the EDL potential turns to be

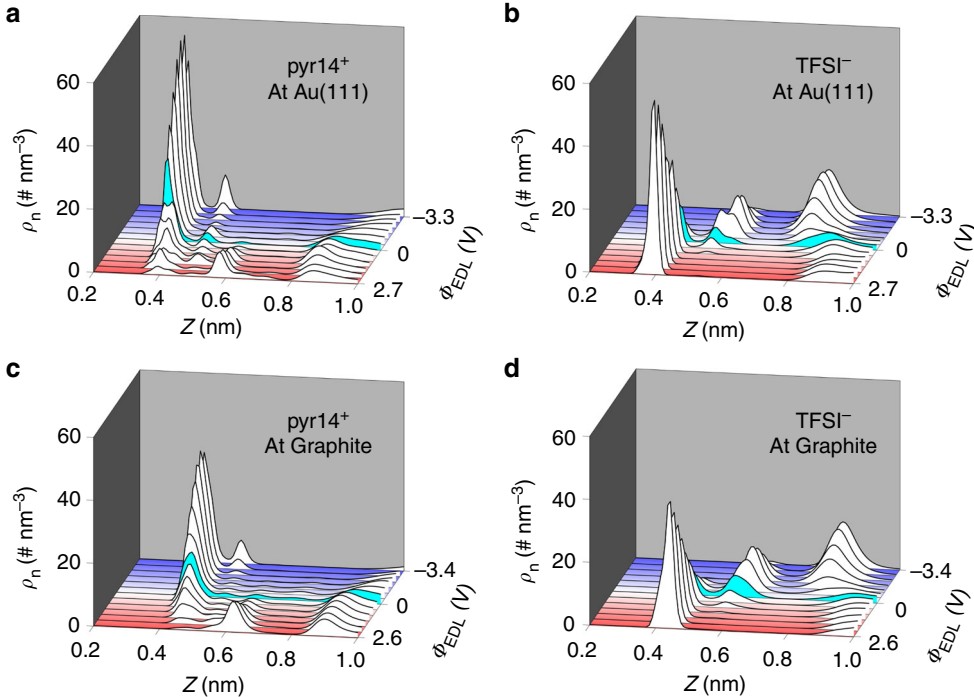

**Fig. 1** Effects of the electrode material on interfacial ion distributions. Ion number density ($\rho_n$) as a function of distance from the electrode ($Z$) under different electrical double layer (EDL) potential ($\Phi_{EDL}$). **a–d** Cation 1-butyl-1-methylpyrrolidinium (pyr14$^+$) (**a**) and anion bis(trifluoromethylsulfonyl)imide (TFSI$^-$) (**b**) number density profiles near gold surface. pyr14$^+$ (**c**) and TFSI$^-$ (**d**) number density profiles near carbon surface. Blue and red colors highlight negative and positive potential biases, respectively. Note that planes across centers of surface atoms of carbon and gold electrodes are located at $Z = 0$ nm, but the results are displayed in the relevant range of 0.2 nm<$Z$<1 nm. Through this work, number density is computed based on the center of mass of each ion

positive, pyr14$^+$ can still be adsorbed at the gold electrode surface even at quite large EDL potential (see Fig. 1a, $\Phi_{EDL} = 2.7$ V), whereas for carbon, the first adsorbed cation layer decays quickly and eventually disappears as $\Phi_{EDL}$ rises (Fig. 1c). Similar observations can be found for RTIL [BMIM][BF$_4$] as well (Supplementary Note 3, Supplementary Figure 4). These phenomena agree well with the neutron reflectometry measurement, in which a cation-rich interface was observed at a positively charged gold electrode, showing that gold brings on specific adsorption of cations[43]. We also compare the ion distribution near gold and carbon electrodes, under the same electrode surface charge density instead of the same EDL potential, and reach the same conclusion (Supplementary Note 3, Supplementary Figure 5).

Much to our surprise, although gold and carbon electrodes lead to quite different interfacial ion distributions, MD simulations reveal that the influence of the types of electrodes on water electrosorption behavior is limited. Panels a and b of Fig. 2 exhibit water electrosorption with respect to the EDL potential at the humid [pyr14][TFSI]/electrode interface and humid [BMIM][BF$_4$]/electrode interface, respectively. We focus herein on the first layer of water since such water could be contact-adsorbed on the electrode surface (Supplementary Figure 6). Specifically, we show the number density of water in the "interfacial region" averaged within ~0.35 nm from the electrode, scaled to that of water in the bulk region (i.e., the region beyond EDL region). For each of the studied RTILs, gold and carbon electrodes show very similar trends in water electrosorption as a function of the EDL potential.

The hydrophobic/hydrophilic nature of RTILs is, however, very important. It substantially affects the absorbed water distribution. For the hydrophobic RTIL [pyr14][TFSI] (Fig. 2a), it can be found that: (1) The water in the interfacial region gets enhanced in general as the absolute value of the EDL potential increases,

regardless of the types of electrodes. (2) The electrosorption of water is stronger near positively charged electrodes than near negatively charged electrodes under a wide range of the electrode potential.

For the hydrophilic RTIL [BMIM][BF$_4$], the shape of water electrosorption curves is very different from the hydrophobic RTIL [pyr14][TFSI]. That is, under negative polarization, water incorporated in [BMIM][BF$_4$] will not be adsorbed on either two types of electrodes. Note that in our simulations, water content for [BMIM][BF$_4$] is about two times as much as that for [pyr14][TFSI] (4953 and 2796 ppm for [BMIM][BF$_4$] and [pyr14][TFSI], respectively, see Supplementary Table 1). These results indicate that hydrophilic RTILs absorb more water in the bulk, they, however, would not let it move to negatively charged electrode and get involved into electrosorption. The nature of this effect is rationalized in the next section.

Additional simulations on these two studied RTILs with different water contents reveal that the influence of water concentration on the observations in Fig. 2 is weak (Supplementary Note 4, Supplementary Figure 7). Even under ~23,000 ppm of water in humid RTIL [BMIM][BF$_4$], water depletion could still be observed at negatively charged electrode (Supplementary Figure 7b).

To validate the MD-revealed effects that the water accumulating at the interface in hydrophobic RTIL [pyr14][TFSI] would increase with higher electrode polarization of any sign, whereas water in hydrophilic RTIL [BMIM][BF$_4$] would rarely contact a negatively polarized electrode, we carried out CV measurements for both [pyr14][TFSI] and [BMIM][BF$_4$] on Au(111) electrodes. Prior to CV measurements, to check that we deal with atomically flat surface so that surface roughness would not complicate our conclusions, AFM characterization was performed and verified this (Supplementary Note 5, Supplementary Figure 9).

For humid RTIL [pyr14][TFSI] (Fig. 3a), when water content is increased from 538 to 1260 ppm, the electrochemical window is narrowed to some extent; with water content enlarged from 1260 to 3815 ppm, the decrease of electrochemical window was observed distinctly: the lower limit of electrochemical window gets moved from around −2.4 to −1.9 V. Since all experimental conditions except water content was kept unchanged, it is reasonable to attribute the decrease of electrochemical window to the reduction of water. When potential scans positively, the characteristic peaks are present. Taking 1260 ppm water content as an example, an anodic current wave at around +2.0 V and a cathodic peak at about +1.4 V were observed, which stand for the oxidation of Au(111) surface and the corresponding reduction, respectively.

On the contrary, for humid RTIL [BMIM][BF4], CV characteristics are similar as water content changes from 14 to 3014 and then to 22,800 ppm. As shown in Fig. 3b, electrochemical windows remain nearly unchanged even considering the remarkable change of water content.

The above results demonstrate that water molecules in hydrophobic RTIL [pyr14][TFSI] eagerly adsorb onto the electrified electrode surface and thus favor the electrochemical reaction of water. In contrast, water molecules in hydrophilic RTIL [BMIM][BF4] prefer keeping a distance away from the negatively polarized electrode surface, which could hinder the electrochemical reduction of water. CV measurements on highly oriented pyrolytic graphite (HOPG) electrodes were also investigated to explore the impact of electrode type, which exhibit the same trend as seen for Au(111) electrode concerning the effect of water electrosorption on electrochemical activity in hydrophobic and hydrophilic RTILs (Supplementary Note 4, Supplementary Figure 8).

**Origin of water electrosorption**. To delve deeper into above observations based on different electrode materials and RTILs, we evaluated the mean force potential of a water molecule as a function of the distance to the electrode surface under different EDL potentials. Firstly, by use of umbrella sampling method[44], we computed the potential of mean force (PMF) for water. Exactly, it represents the variation of free energy with changing the position of the center of one water molecule from the electrode surface towards the bulk RTIL region, averaged over orientations of water molecules and lateral positions along the electrode. We derive the PMF for water in [pyr14][TFSI] and [BMIM][BF4] at gold electrode when the potential applied between the two electrodes is 4 V (Fig. 4, also see Supplementary Note 6 and Supplementary Figure 10 for an applied potential of 2 V). The results for carbon electrode are presented in Supplementary Figure 11 (Supplementary Note 6), as they are qualitatively similar to those in Fig. 4.

For all systems, one could see that near both positive and negative electrodes, the positions of global and local minima of PMF curves correspond quite well with the water number density profiles. The hydrophobic RTIL [pyr14][TFSI] leads to a well-defined minima of PMF at both positive and negative electrodes, but the potential well near the negative electrode (−0.097 eV at ~0.25 nm, Fig. 4a) is less deep than the one near the positive electrode (−0.144 eV at ~0.26 nm, Fig. 4b).

However, the situation is different in the hydrophilic RTIL [BMIM][BF4]. Herein, we find a positive potential well near the negative electrode (+0.014 eV at ~0.26 nm, Fig. 4c). This can cause only a metastable adsorption, because its minimum lies higher than the free energy of water in the bulk (which we conventionally assume at distances larger than ~2.0 nm from each electrode, as PMF decays there to zero). Near the positively charged electrode, there is a negative minimum near the interface (−0.089 eV at ~0.26 nm, Fig. 4d), similar to that of [pyr14][TFSI]. These PMF profiles confirm the conclusion that in strongly polarized cells, water molecules in hydrophobic RTILs prefer to accumulate at the surfaces of both electrodes, while water molecules in hydrophilic RTILs are reluctant to stay near the negative electrodes.

As PMFs demonstrate where water would be predominantly distributed, we then focus on the structures of water within the interfacial region. Since water in hydrophobic and hydrophilic RTILs undergoes different association with cations/anions, one can expect that the structures of interfacial water may depend on the nature of the RTILs, EDL potential, and electrode type. Revealed by the orientations of water in [pyr14][TFSI] and [BMIM][BF4] at the electrified electrode (Supplementary Note 7, Supplementary Figure 12), it could be concluded that (1) the interfacial water tends to orient vertically to the negative electrode but lies flat on the positive electrode; (2) for negatively charged electrode water orientation in different RTILs exhibits some differences, but no notable differences are seen at the positively charged electrode. Since the surface charges would always force the dipole of water to align vertically to the electrode surface, the

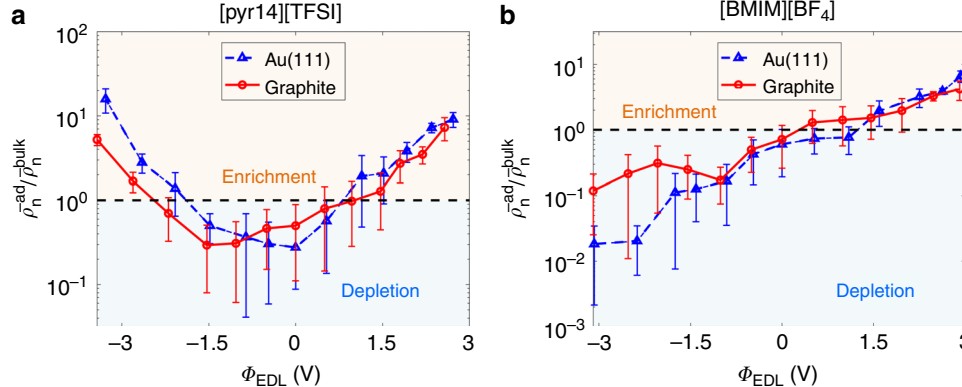

**Fig. 2** Electrosorption of water from humid ionic liquids on electrodes. **a, b** show water adsorption from the humid 1-butyl-1-methylpyrrolidinium bis (trifluoromethylsulfonyl)imide ([pyr14][TFSI]) and 1-butyl-3-methylimidazolium tetrafluoroborate ([BMIM][BF4]) with respect to the electrical double layer potential ($\Phi_{EDL}$), respectively; results for gold electrode are shown by blue triangles and for carbon electrode by red circles. The relative enrichment or depletion of water molecules in the first interfacial layer is depicted by the ratio of averaged number density of water in this layer to that of water in the bulk region ($\bar{\rho}_n^{ad}/\bar{\rho}_n^{bulk}$). Enrichment (depletion) zone corresponds to a higher (lower) water density than in the bulk. The error bars are due to the standard deviation of the results based on three independent simulation cases. Source data are provided as a Source Data file

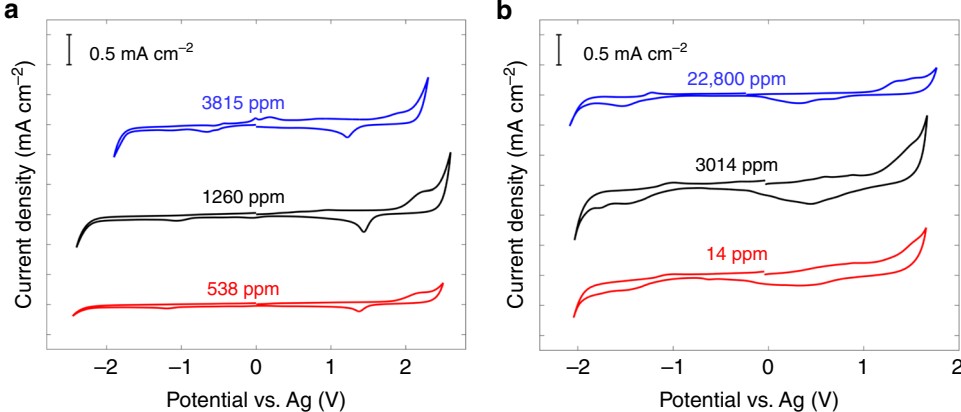

**Fig. 3** Effects of water electrosorption on electrochemical activity on gold electrode. **a**, **b** Cyclic voltammograms of Au(111) in hydrophobic 1-butyl-1-methylpyrrolidinium bis(trifluoromethylsulfonyl)imide ([pyr14][TFSI]) (**a**) and hydrophilic 1-butyl-3-methylimidazolium tetrafluoroborate ([BMIM][BF$_4$]) (**b**) under different water contents. Scan rate: 100 mV s$^{-1}$

discrepancy of water orientation at cathode and anode would be attributed to the different water–RTIL associations, to a large extent, determined by water–ion hydrogen bonding. Detailed analysis of hydrogen bonding between water and cations/anions in the interfacial region, which can be found in Supplementary Note 7 (Supplementary Figure 13), unveils that (1) water forms more H-bonds with anions than cations, even in the cation-rich EDL under negative polarization, suggesting that anions play a dominant role in hydrophobicity/hydrophilicity of RTILs; (2) more water–ion H-bonds are observed in hydrophilic [BMIM][BF$_4$] than in hydrophobic [pyr14][TFSI], consistent with previous work[45,46].

The overall water–RTIL hydrogen bonding as well as water orientation strongly depends on the potential drop across the EDL. Thus, naturally, the electrosorption of water depends not only on the water–ion interaction, "water following anions", but also on the water–electrode interaction.

In order to further quantitatively explore the interaction energy of an interfacial water molecule with its nearby species at the polarized electrode, as a function of EDL potential, and more importantly, to distinguish the contributions from water–electrode and water–ions interactions, herein the total interaction energy is divided into four components: (1) water–ions van der Waals interaction, (2) water–ions coulombic interaction, (3) water–electrode van der Waals interaction, and (4) water–electrode coulombic interaction. The long-range coulombic interactions of water with electrodes and electrolyte ions are accurately evaluated by the modified partial-mesh Ewald (PME) approach (described in Methods section); the short-range van der Waals interactions are calculated based on Lennard–Jones potential by cutoff summation[35]. Figure 5 shows the interaction energy with its components of a water molecule in hydrophobic RTIL [pyr14][TFSI] and hydrophilic RTIL [BMIM][BF$_4$] within the electrosorption layer for different EDL potential, on gold and carbon electrodes.

For all systems studied, the total interaction energy as a function of EDL potential (magenta lines) could account well for the water electrosorption depicted in Fig. 2. Focusing on its four components, one can notice that two components, water–ions coulombic interaction and water–electrode coulombic interaction, contribute the most to the change of total potential energy than the other two (water–ions van der Waals interaction and water–electrode van der Waals interaction). For instance, for both studied RTILs and electrodes, the water–electrode coulombic interaction increases drastically as the EDL potential becomes more positive or negative, while the water–electrode van der

Waals interaction, as expected, varies very little within a wide range of EDL potential.

Based on the observation in Fig. 2, electrode seems to play a minor role in water electrosorption on the charged surfaces. Why is that? In fact, gold electrode does lead to much stronger van der Waals interaction with water than carbon (its water–gold value is almost twice as much as the value for water–carbon, with around −0.127 and −0.068 eV, respectively, see blue bar charts in Fig. 5). However, as the EDL potential increases (or decreases), water–electrode coulombic interaction becomes an ever-growing share of the total interaction energy. Especially when the electrode is highly charged, the water–electrode coulombic interaction will dominate over the water–electrode van der Waals interaction, which attenuates the effect of types of electrodes on water electrosorption.

This component analysis of interaction energy effects on water distribution can further instruct us, why water would "feel itself less comfortable" at the [BMIM][BF$_4$]/electrode interface under negative polarization than in the bulk region. For [BMIM][BF$_4$], a monotonous trend of total interaction energy of water with respect to EDL potential (magenta dots in Fig. 5b, d) could be seen to differ a lot from the bell-shaped trend of the curves for [pyr14][TFSI] (magenta dots in Fig. 5a, c), regardless of electrode materials. This can be attributed to several factors:

(1) The miscibility of an RTIL with water is mostly determined by the associated anion[12,47]. Under negative polarizations, as anions are pushed away from electrode surfaces, water in [BMIM][BF$_4$] would have less propensity to be close to the electrode. This effect will be much less pronounced in [pyr14][TFSI] owing to the weaker hydrophilicity of TFSI⁻ anion.

(2) Both van der Waals interactions of water with ions and electrode fluctuate a little; however, the coulombic interactions change a lot as applied potential varies. This makes the van der Waals interactions negligible because they contribute little to the change of total interaction energy.

(3) Although water–ions interaction is weakened near negatively charged electrodes, water–electrode coulombic interaction would enlarge, due to the electrostatic attraction between charges on electrodes and water (as a dipole).

Take gold electrode as examples, for [BMIM][BF$_4$], the water gets weaker coulombic interactions with electrolyte ions by 0.917 eV (from −0.565 eV at $\Phi_{EDL}$ = 0 V to +0.352 eV at $\Phi_{EDL}$ = −3.28 V); meanwhile, the water–electrode coulombic interaction is enhanced by 0.652 eV. Hence, at $\Phi_{EDL}$ = −3.28 V the total interaction energy

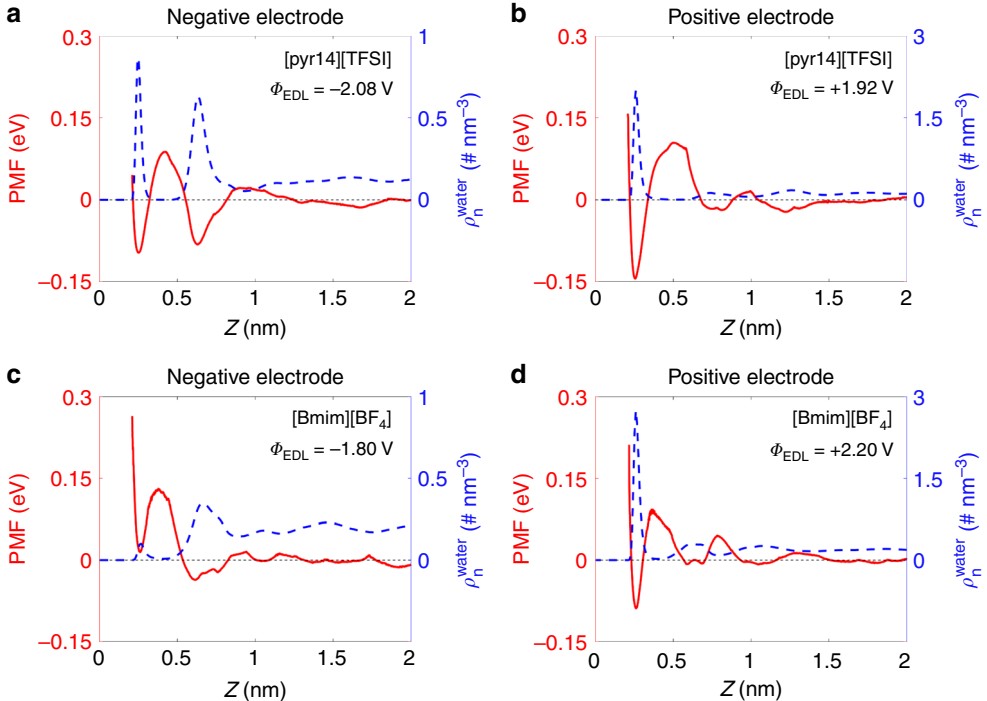

**Fig. 4** The propensity for water electrosorption at gold electrodes. **a–d** Correlation between the distributions of water density (dashed blue lines, right blue y-axis) and the potential of mean force (PMF) for water (solid red lines, left red y-axis) for room temperature ionic liquids at gold electrodes: in 1-butyl-1-methylpyrrolidinium bis(trifluoromethylsulfonyl)imide ([pyr14][TFSI]) at **a** negative and **b** positive electrodes, and in 1-butyl-3-methylimidazolium tetrafluoroborate ([BMIM][BF$_4$]) near **c** negative and **d** positive electrodes. The applied potential between the two electrodes is 4 V, and the PMF curves are obtained by umbrella sampling along the distance from the electrode surface. The electrical double layer potential ($\Phi_{EDL}$) at negative or positive electrode is labeled in each panel

of water with the environment is actually weakened compared with that at $\Phi_{EDL} = 0$ V, which leads to the depletion of water. Whereas for [pyr14][TFSI] under similar condition, the water–ions coulombic interaction is changed by 0.603 eV (from −0.336 to +0.267 eV) and water–electrode coulombic integration is strengthened by 0.753 eV, resulting in a larger total interaction energy. That is to say, although electrified electrodes can attract water, at negative polarization the overall hydrophilicity of [BMIM][BF$_4$] becomes irrelevant in the cation-rich double layer, so that more water molecules prefer to be absorbed in the bulk rather than at the negatively charged electrode.

**Generality based on different RTILs.** How general is the conclusion that the electrochemical window of hydrophilic RTILs is "waterproof"? To answer this question, MD simulations and CV measurements are both carried out for another two RTILs: [BMIM][TFSI] and BMIM-trifluoromethanesulfonate ([BMIM][OTf]); the former is hydrophobic and the latter is hydrophilic[48–50].

Although cations pyr14$^+$ and BMIM$^+$ are very different in terms of their molecular structures, the observations in hydrophobic [BMIM][TFSI] (Supplementary Figures 14a and 15a), both in MD simulations and experiments (Supplementary Note 8), follow the same trend uncovered for hydrophobic [pyr14][TFSI]. On the other hand, similar to hydrophilic [BMIM][BF$_4$], one can see that water in hydrophilic [BMIM][OTf] stays away from the negative electrodes (Supplementary Figure 14b), so that the electrochemical window could be maintained with the increase of water content (Supplementary Figure 15b). This fact is very important to know if one wants to use RTILs as electrolytes in supercapacitors or as a solvent for electrocatalysis.

The comparison of three RTILs studied, [pyr14][TFSI], [BMIM][TFSI] and [BMIM][BF$_4$], in terms of the interfacial water orientation and hydrogen bonding with RTIL ions, discloses that it is anion type (which dominates the hydrophilicity of RTIL) rather than cation type that majorly determines the enrichment or depletion of water electrosorption at the negative electrode (Supplementary Note 8, Supplementary Figure 16). Indeed, more RTILs should be tested to prove the universality of this effect, but its physical nature seems to be very sound.

## Discussion
We have studied the impact of different electrode materials and hygroscopicity of RTILs on interfacial distribution of ions and electrosorbed water, using MD simulations and CV measurements. Our simulations have revealed that for the two major studied RTILs, gold electrode has much stronger ion adsorption than carbon electrode, resulting in a quite different electrical double layer structure. For instance, cations can be adsorbed on the gold electrode surface even under moderate positive polarization. However, the type of the electrode does not qualitatively change the feature of water electrosorption on electrodes from a humidified RTIL. Instead, the hygroscopic nature of RTILs and the specific interactions of water with its cations and anions play a key role, so that for negative electrode polarization, water in hydrophilic RTIL is much less adsorbed on the electrode surface than in hydrophobic RTIL. The latter is verified by our experimental observations, which show that a hydrophilic RTIL leads to less electrochemical reduction even though it typically has higher water content, if absorbed from humid air.

These observations can be well explained by energetics of water interaction with the environment (RTIL and electrode). First, the PMF as a function of the distance from the electrode uncovered

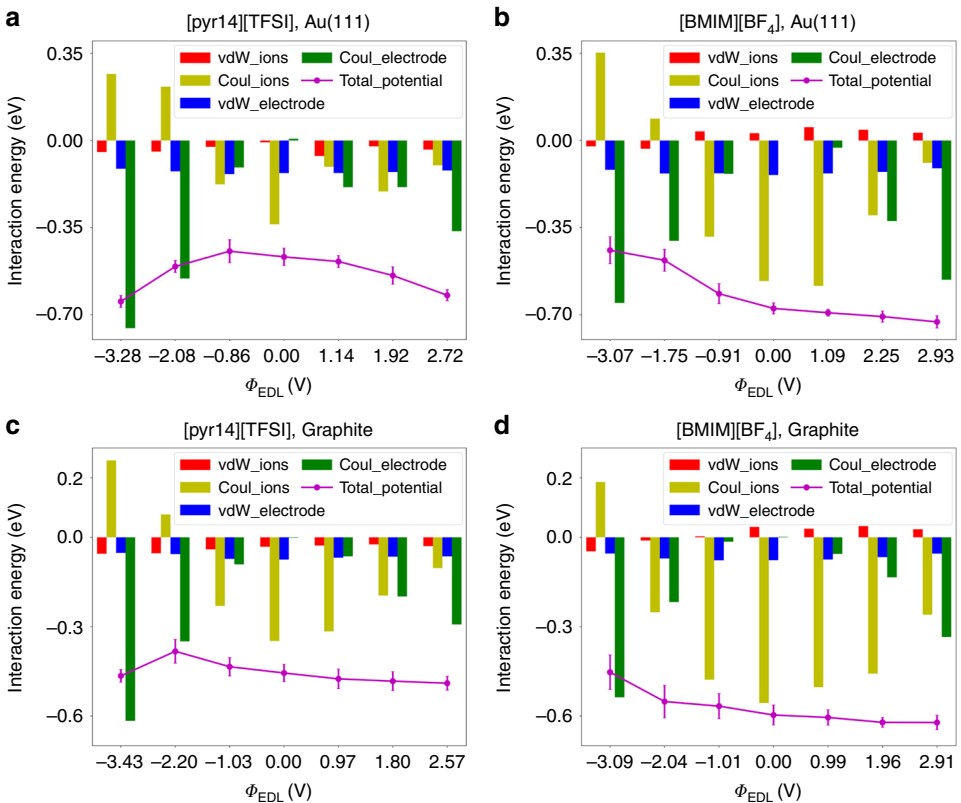

**Fig. 5** Interaction energies at different electrode polarizations. The weight of different components of the total interaction energy of water molecule adsorbed at the electrode surface under different electrode polarizations ($\Phi_{EDL}$). **a–d** The figures for water adsorbed from 1-butyl-1-methylpyrrolidinium bis (trifluoromethylsulfonyl)imide ([pyr14][TFSI]) (**a**) and 1-butyl-3-methylimidazolium tetrafluoroborate ([BMIM][BF$_4$]) (**b**) near gold electrode and from [pyr14][TFSI] (**c**) and [BMIM][BF$_4$] (**d**) near carbon electrode. Red, yellow, blue, and green bar charts represent water–ions van der Waals, water–ions coulombic, water–electrode van der Waals, and water–electrode coulombic interactions, respectively. The magenta dots connected by solid lines are the total value of "local" potential energy of water with its nearby species in the interfacial region, which results from the summation of previous four interactions. The error bars for magenta dots are due to the standard deviation of the results for three independent simulation cases. Source data are provided as a Source Data file

that water within the interfacial region has a negative free energy, causing an accumulation of water near electrode surfaces, except for water incorporated in hydrophilic RTILs at the negatively charged surface where it has a positive free energy. Further, accurately analyzing the weight of four kinds of mutual interactions: (1) water–ions van der Waals interaction, (2) water–ions coulombic interaction, (3) water–electrode van der Waals interaction, and (4) water–electrode coulombic interaction, we could justify a relatively weak effect of the nature of the electrode on water electrosorption and the drastically weakened interaction of water incorporated in hydrophilic RTILs with its surroundings in the cation-rich double layer near negative polarization.

The findings reported here could provide guidelines for minimization of water electrosorption. It is based on, at a first glance, a counterintuitive conclusion from this study that using more hydrophilic RTILs could actually keep water away from the negative electrode. Since water would not accumulate near negatively electrified electrode, the wide chemical window may be well exploited without water splitting occurring at negative polarization.

Although the generality of this conclusion is supported by the control studies with two other RTILs (one hydrophobic and one hydrophilic), further work is still required (both simulation-wise and experimental), as different RTILs have various degrees of hydrophilicity or hydrophobicity. Moreover, seeking new approaches that could reduce water electrosorption at both positive and negative electrodes requires ingenuity, and would be

challenging but in high demand for various electrochemical applications.

## Methods

**MD simulations.** The schematic of MD simulation setup is shown in Supplementary Figure 1 as a channel system. In total, four major systems (combinations of the two types of electrodes and two different RTILs) as well as two control modeling systems (another two sets of hydrophobic/hydrophilic RTILs on gold electrodes) are considered in this work. We model gold and carbon by Au(111) single-crystal surfaces and three layers of graphene sheets, using the force fields from Halicioǧlu and Pound [51] and force fields from Cornell et al.[52], respectively. RTILs [pyr14][TFSI] and [BMIM][BF$_4$], along with RTILs [BMIM][TFSI] and [BMIM][OTf], are modeled by all-atom force fields[53–55], which provide the thermodynamic and dynamic properties very close to the experimental data. The TIP3P model is used for water molecules. In theory, the electronic polarizability of water molecules and RTIL ions should be accounted for accurately characterizing their mutual interactions and dynamics. However, previous simulations based on non-polarizable water/RTIL models have successfully captured key features of the water–RTIL mixture[46,56,57]. Therefore, the classical force fields adopted here are deemed adequate. The sizes of all simulation cells are chosen as large enough to reproduce the bulk state of the RTILs in the central region between two electrodes. Detailed number of the components and water contents are gathered in Supplementary Table 1 for all studied systems.

**Constant potential method.** In order to capture the electrode polarization effects in the presence of electrolytes, constant potential method (CPM) is implemented in a methodology developed by Raghunathan and Aluru[58], as it directly applies a constant electrode potential by allowing fluctuations of charge on electrode surface during the simulation[28,59,60]. This method is in good agreement with other implementations of CPM[61–63], and specialized optimization and parallelization are implemented to make the additional computation cost rather limited. It is worth to

note that CPM by far can mainly focus on the electrolyte side of the interface and cannot adequately describe the potential distribution in the electrode (electrode is modeled as a conductor). Therefore, the levels of "metallicity" of gold and carbon electrodes are not taken into account in this work. All simulations are performed in NVT ensemble, using a customized MD code Gromacs[64], with a time step of 2 fs. The electrolyte temperature is maintained at 338 K, using the Nosé–Hoover thermostat. The electrostatic interactions are computed using the PME method and by solving an auxiliary Laplace equation[58]. An FFT (fast Fourier transform) grid spacing of 0.1 nm and cubic interpolation for charge distribution are used to compute the electrostatic interactions in the reciprocal space. A cutoff length of 1.2 nm is chosen in the calculation of electrostatic interactions in the real space. The non-electrostatic interactions are computed by direct summation with a cutoff length of 1.2 nm. For each simulation, the MD system is first annealed from 600 to 338 K over a period of 10 ns, following by running another 40 ns to reach equilibrium. Then, a 150-ns production run is performed for analysis. For each case, three independent simulations were run with different initial configurations to ensure the accuracy of the modeling results.

**Interaction energy analysis**. We characterize the local (non-mean-field) interactions between water molecules and their surroundings in an EDL based on MD-obtained trajectories. Such analysis has been widely used in prior simulation studies[35,65]. The algorithm adopted herein is upgraded by performing PME method instead of inaccurate cutoff method, to precisely and separately compute the long-range electrostatic interactions of water with the electrodes and with surrounding electrolyte ions.

**Experimental materials and electrochemical measurements**. RTILs [pyr14] [TFSI] and [BMIM][TFSI] were purchased from IoLiTec in the highest available quality (99%). [BMIM][BF$_4$] and [BMIM][OTf] were purchased from Lanzhou Yulu Fine Chemical Co., Ltd in 99% quality as well. Prior to each measurement, the RTILs were vacuum-dried several hours at 80 °C in a glovebox filled with ultra-pure Ar (99.999%) to remove as much absorbed water as possible, and then used as dry RTILs. To increase the water concentration in the RTILs, more water was added into 1000 μL RTILs and stirred, and then used as humid RTILs. Water concentration was measured by using the Karl Fischer method[66,67] by using an 831 KF Coulometer (Metrohm). Au(111) single-crystal electrodes (Supplementary Figure 9) were made by using the Clavilier method and were subjected to electrochemical polishing and flame annealing in hydrogen followed by cooling under a nitrogen atmosphere prior to each experiment. HOPG was purchased from Bruker. A piece of tape was pressed onto the flat surface and then pulled it off. The freshly cleaved surface of HOPG was used for measurements. CV measurements were carried out by using an Autolab electrochemical workstation (Eco Chemie, The Netherlands) in a glovebox. A silver wire was used as the reference electrode, and a platinum wire was used as an auxiliary electrode. The electrochemical cell used for measurements was a sealed one, which was isolated from the environment, and CV measurement was typically completed within half an hour, so that the water concentration level was kept stable.

## Data availability
The data that support the findings of this study are available from the corresponding authors upon request.

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

## Acknowledgements

G.F., S.B., and R.X.W. acknowledge the funding support from the National Natural Science Foundation of China (51876072) and Shenzhen Basic Research Project (JCYJ20170307171511292). B.W.M., J.W.Y., and S.L. acknowledge the support of Natural Science Foundation of China (21673193, 21533006, 21621091, and 21727807). A.A.K. acknowledges the support from The Leverhulme Trust Research Grant (PRG-2016-223) and HUST for the support of this project through the HUST Advisory Professorship and Imperial College for the support of this form of collaboration between the involved HUST and Imperial groups. The computation is completed using Tianhe II supercomputer in National Supercomputing Center in Guangzhou.

## Author contributions

G.F. conceived this research. G.F. and J.W.Y. designed the work of simulation and experiment, respectively. S.B. and R.X.W. carried out all simulations. S.L., J.W.Y., and B. W.M. carried out the experiment. G.F., S.B., J.W.Y., and A.A.K. wrote the manuscript. All authors contributed to the data analysis and commented on the manuscript.

## Additional information

**Competing interests:** The authors declare no competing interests.

