## [Peer Review File · Nature Communications]

Reviewers' Comments:

Reviewer #1:

Remarks to the Author:

In this manuscript Bi et al. report a combined simulation and experimental study in order to characterize the adsorption of water at the surface of electrodes, when this molecule is not the solvent but rather an impurity inside an ionic liquid. This is a very timely topic since many groups are currently investigating electrolytes with low water concentrations, such as for example "water-in-salts". The findings are very interesting (not only for energy storage applications, but for a broad range of problems in electrochemistry and in catalysis), the study is well conducted and the paper is well written, so that I strongly recommend publication in Nature Communications. However I think the authors should address the following points prior to publication:

1/ From the technical point of view, I am only worried by the accuracy of the comparison between the two electrode types. In fact, gold and carbon electrodes just differ by their structure and their van der Waals parameters in the simulations. However, it is well known that they have quite different electronic structure, i.e. levels of "metallicity". I think that the authors use state-of-the-art methods to simulate metallic structure, so that it is not possible to go much further, but this limitation should clearly be stated for non-specialist readers.

2/ The main objective of the paper is to study the reactivity of the water at the interface. In my opinion, this reactivity may strongly be affected by the orientation of the water molecules (i.e., are the proton or the lone pairs more likely to be involved in a reaction occurring at the surface), so I think it would be very interesting if the authors could provide a detailed analysis of the orientation of the water molecules at the interface with respect to the electrode type, nature of the ionic liquid and applied potential. They should have all the material to do this analysis at hand, so I do not think this would require much work from them (unless some interesting behaviors are observed and need to be further understood!)

3/ Even if there is only one graphene layer in their electrode, I am not sure that real "graphene" would behave as shown in this study. Due to the accessibility of the two faces of the plane, peculiar adsorption behaviors have recently been shown in molecular dynamics studies (both for water and for ionic liquids). I think the material simulated by the authors is somewhat more similar to graphite, so that I recommend to switch graphene to graphite in the figures and in the text.

4/ Why didn't the authors perform similar experiments on a carbon electrode? Here they only did gold, but being able to do the same comparison as in the simulations would strengthen the conclusions.

5/ I do not agree with the statement line 214. The PMFs shown on Figure 4 do not correspond to a simple "potential energy". This should be corrected.

6/ Minor point: Line 379 please correct "inasmuch"

Reviewer #2:

Remarks to the Author:

The manuscript "Reducing water electrosorption to electrodes from humid ionic liquids" focusses on the electrosorption behavior of water in ionic liquids on two different surfaces (gold and graphene). The authors use two different ionic liquids (one hydrophilic and one hydrophobic) and study water adsorption behavior by complementary molecular dynamic (MD) simulation and cyclic voltammetry

(CV). The authors conclude that water will be depleted on negatively charged electrodes in a hydrophilic ionic liquid system in general.

While the presented work appears to be very well done, I feel that the authors do not have enough data and control experiments to support a general conclusion that warrants publication in a journal such as Nat. Comm. Currently, the data is good enough for a more specialized outlet after major revisions. Hence, I feel that the work is currently premature for consideration in Nat. Comm. and generally not ready for acceptance. However, considering the potential interest in ionic liquids (and the effect of water in ILs), a substantially revised version with appropriate control experiments may be reconsidered for publication in Nat. Comm. My major concerns are as follows:

1) My first doubt about generalization is the specific water level used: Is there any particular reason for using about 3000 ppm of water in both ILs? Is this possibly the maximum which the hydrophobic IL can absorb? I can see that one wants to compare similar levels. Yet, what would be the effect of a higher concentration (if possible). Or more general: Is there a concentration dependence of the observed effect?

2) Mentioning the maximum water content at room temperature will be interesting. In particular, considering that the authors label their ILs hydrophilic and hydrophobic. What is the criterion for this label? Do they mean water soluble vs. not water soluble? Or do they refer to the ability to take up water above a certain threshold?

3) The authors state that the hydrophilicity of the ionic liquid is responsible for the observed behavior. However, the so-called "hydrophobic/hydrophilic" character of the IL itself (however the authors wish to define this) may not be the correct descriptor for the observed behavior. I am not yet convinced, and as the devil's advocate I argue that the actual molecular structure of the cations plays a more central role in this case. Specifically, BMIm⁺ has planar delocalized charge (π orbital delocalization) and pyr14⁺ has a point charge on a tertiary ammonium, which certainly leads to different interaction/packing behavior at the surface and to different interactions with water itself. As such, it seems equally likely that the observed effects are cation specific, rather than being connected to the hydrophilicity of the liquid itself.

4) In the same direction, the role of the anion is not discussed at all, even though the anions are in this case partly responsible for the overall hydrophobic/hydrophilic character of the liquids. The authors compare two ILs with two cations AND two different anions (a hydrophilic and a very hydrophobic anion). Hence, the overall character of the liquids is equally affected by the anions. As such, making a generalizing statement utilizing the overall liquid character seems very weakly supported by evidence at this point.

5) Hence, I feel that the authors must provide proper control experiments with both cations and TFSI and/or BF₄ as a common anion. As additional control, the authors will be able to further generalize by using e.g. a similar cation with TFSI and e.g. the very hydrophilic Cl⁻ anion, allowing to relate the behavior more convincingly to an overall solution behavior.

6) Also, in figure 1 and supplementary figure 2, the authors show MD derived polarization dependent ion distributions at electrode/IL interfaces. The correlation/interaction between water molecules and ions is not well discussed at all but also available from these simulations. Again, the structure of the ions is very different, and water interactions will certainly be different for different ions. How this correlates with the observations is unclear. I can imagine that some of the water is bound strongly to some of the anions and/or cations, which in turn of course changes the amount of overall available water for interfacial adsorption, or this may even result in co-adsorption. Again, this suggests that one has to rule out or at least understand potential water concentration (and threshold) effects, in order to support a generalizing conclusion (see again bullet point 1).

7) Figure 3b: The overall current level is about 10x less compared to the 3a. This should be emphasized. My question here is: Why is the current on the anodic side so much lower as well. The argument of the authors is, that the hydrophilic IL suppresses the cathodic water adsorption but not

the anodic side (at least in the simulations). The anodic side seems similar in simulations. Hence I would expect a similar current and significant gold oxidation/reduction in experiments as well. This is not observed. Why?

Minor aspects:

1) In figure 1, the potential axis is a bit misleading, 1b and 1d go from negative to positive, while 1a and 1c go from positive to negative. Please emphasize the inversion of the axis orientation in the figure chart and/or caption.

2) According to the experiment procedure, all experiments were conducted in an Ar-filled glovebox. Humid ionic liquids are prepared with mixing dry ionic liquid with water. Did the authors measure the water evaporation rate from the ionic liquid system into the glovebox? In previous work by other authors, the amount of water in ionic liquid systems is usually equilibrated with the surrounding humidity, which allows to generate stable values. Here, water levels in the IL may drop/ change with time. Such effects should be considered and ideally also measured/ estimated.

Reviewer #3:

Remarks to the Author:

This is an interesting MD study (coupled with some experimental work) on an important problem of water adsorption/desorption on electrified surfaces in the presence of ionic liquid electrolytes. The authors found that the hydrophilic ionic liquid [BMIM][BF₄] leads to water depletion at the negative electrode as the magnitude of its electrode potential increases while the hydrophobic ionic liquid [pyr14][TFSI] results in the enhancement of water density at the electrode surface irrespective of its polarization. This is in agreement with their experimental observation that water is not reduced in [BMIM][BF₄] at the negative electrode. I have following comments that I suggest the authors should consider before the manuscript becomes acceptable for publication:

(1) I suggest that error bars be calculated and displayed in Figures 2 and 5. It is known that the convergence of ionic liquid simulations is very slow, but the authors appear to have decent MD statistics. Therefore the presentation of error bars will strengthen the quality of their analysis and give the reader good perspective on the authors' MD results.

(2) I wonder how the electronic polarizability of water and ionic liquids will influence the results, in particular, the interaction energies in Figure 5. I could be wrong but I suspect that water at the electrode surface would be affected most. Furthermore, its effect would probably increase with the magnitude of the electrode potential. Some comments are warranted on this issue.

We've studied all the comments carefully and made point-to-point response to each comment. Specifically, reviewers' comments are copied in blue, and each comment is followed by our response in black. The manuscript has been revised, accordingly, in red. After each response to comment, a brief summary is provided of what has been changed or added and where they are positioned in the revised manuscript and corresponding Supplementary Information.

Here, we briefly summarized the work done for this revision:

- 1) Analysis of the orientation and hydrogen bonding of water molecules at the interface from molecular dynamics (MD) simulation.
- 2) Experimentation and MD simulation on a carbon electrode (HOPG) with hydrophobic [pyr14][TFSI] and hydrophilic [BMIM][BF₄] to show the effect of the nature of the electrode.
- 3) Control experiments on different hydrophobic RTILs ([pyr14][TFSI], [BMIM][TFSI]) and hydrophilic ([BMIM][BF₄], [BMIM][OTf]) RTILs at gold electrodes with various water concentrations to examine impacts from ion type and water concentration, in which [BMIM][TFSI] and [BMIM][OTf] are the new RTILs, not considered in the previous version of our manuscript, chosen for comparison.
- 4) The corresponding new MD simulations were performed for comparison with new experiments. Specifically, MD simulations of different hydrophobic ([pyr14][TFSI], [BMIM][TFSI]) and hydrophilic ([BMIM][BF₄], [BMIM][OTf]) at gold electrodes with various water concentrations to examine impacts from ion type and water concentration.
- 5) MD simulations and experiments of humid RTILs with different water contents have been carried out to investigate the influences from water concentration and electrode type.
- 6) New MD simulations and experiments on hydrophobic [pyr14][TFSI] and hydrophilic [BMIM][BF₄] have been performed to explore the character of hydrophobicity, hydrophilicity and hygroscopicity of RTILs.
- 7) Detailed discussions as well as the revisions have been made based on Reviewer #3's reservations about the account of the electronic polarizability of water and ionic liquids and how that may influence the results.

Responses to Reviewer #1's Comments

Reviewer #1 (Remarks to the Author):

In this manuscript Bi et al. report a combined simulation and experimental study in order to characterize the adsorption of water at the surface of electrodes, when this molecule is not the solvent but rather an impurity inside an ionic liquid. This is a very timely topic since many groups are currently investigating electrolytes with low water concentrations, such as for example "water-in-salts". The findings are very interesting (not only for energy storage applications, but for a broad range of problems in electrochemistry and in catalysis), the study is well conducted and the paper is well written, so that I strongly recommend publication in Nature Communications. However I think the authors should address the following points prior to publication:

Response:

We appreciate the reviewer for recognizing the novelty, significance and the potential impact of this work.

Comment 1) From the technical point of view, I am only worried by the accuracy of the comparison between the two electrode types. In fact, gold and carbon electrodes just differ by their structure and their van der Waals parameters in the simulations. However, it is well known that they have quite different electronic structure, i.e. levels of "metallicity". I think that the authors use state-of-the-art methods to simulate metallic structure, so that it is not possible to go much further, but this limitation should clearly be stated for non-specialist readers.

Response:

We thank the reviewer for pointing out the statements about simulating gold and carbon electrodes.

It is true that graphite, glassy carbon, and other types of electrodes like gold have different electronic properties. Gerischer¹ and Yeager² in their classical works have shown that the capacitance of the interface of such electrodes with electrolytic solutions is to a high extent determined by the electronic space charge in the electrode.³ In Ref. 4, the interface of RTILs with graphite was studied, and it was shown that both space charge in the electrode and in the RTIL should be considered on the same footing.⁴ Many papers have tried to describe the contribution of quantum capacitance of low-dimensional carbons,⁵⁻⁸ all showing the important contribution of the electronic response to charging the electrodes.

The work presented in this work operates with effective force-fields under constant electrode potential scheme. Such cutting edge simulations (although still inferior to *ab initio* Carr-Parinello MD simulations⁹), being totally focused on the electrolyte side of the interface, cannot and are not aimed to adequately describe the potential distribution in the electrode. Thus, the conclusion about the competition between water and ions near the electrode surface should be scrutinized by experiments. In the future, they could be tested *in silico* by *ab initio* simulations, which for the system of this size and available computation facilities, would have currently taken astronomic computer time.

The constant potential simulations in this work by far cannot adequately describe the levels of ‘metallicity’ of gold and carbon. However, the gold and carbon (with different electronic structure that is taken into the MD force fields to some extent), taken as electrodes in this work, exhibit similar impact on interfacial water adsorption which is not only revealed by MD simulation but also by the new experiments by using highly oriented pyrolytic graphite (HOPG) as working electrode. The latter has its surface structure very close to the model we used for simulation. The details of the experiments on carbon electrode could be found in responses to **Comment 4** below in which the reviewer suggests to "perform similar experiments on a carbon electrode".

Brief summary:

Based on the reviewer’s advice, the above discussion has been incorporated into the revised manuscript (*Method Section of main text, page 19, lines 450-454*), as:

“It is worthy to note that CPM by far can mainly focus on the electrolyte side of the interface and cannot adequately describe the potential distribution in the electrode (electrode is modeled as a conductor). Therefore, the levels of ‘metallicity’ of gold and carbon electrodes are not taken into account in this work.”

Comment 2) The main objective of the paper is to study the reactivity of the water at the interface. In my opinion, this reactivity may strongly be affected by the orientation of the water molecules (i.e., are the proton or the lone pairs more likely to be involved in a reaction occurring at the surface), so I think it would be very interesting if the authors could provide a detailed analysis of the orientation of the water molecules at the interface with respect to the electrode type, nature of the ionic liquid and applied potential. They should have all the material to do this analysis at hand, so I do not think this would require much work from them (unless some interesting behaviors are observed and need to be further understood!)

Response:

We thank the reviewer for this very constructive comment. We fully agree with reviewer that the orientation of absorbed water molecules would reflect some details on mutual interactions between water and nearby species.

Figure R1.1 | Orientation of interfacial water a-b, Water orientation with hydrophobic [pyr14][TFSI] at gold (a) and carbon (b) electrodes. **c-d**, Water orientation with hydrophilic [BMIM][BF₄] at gold (c) and carbon (d) electrodes. **e**, The cartoon of orientational angles: water dipole orientation, θ_{dipole} , is defined as the angle formed by the water vector and the normal of electrode surface; water normal orientation, θ_{normal} , is defined as the angle formed between the normal of water plane and the normal of electrode surface. Red solid lines show water dipole orientation, and blue dashed lines are for water normal orientation. The EDL potential is labeled on the top of each figure, and the applied voltage between two electrodes is 4V.

Following the reviewer's suggestion, we analyzed the orientation of interfacial water absorbed in these two studied RTILs (hydrophobic [pyr14][TFSI] and hydrophilic [BMIM][BF₄]) on gold and carbon electrodes *via* MD simulation (see **Figure R1.1**). Specifically, as shown in the lowest panel (**Figure R1.1e**), two types of orientational angles – water dipole orientation (θ_{dipole} , the angle formed by the water dipole vector and the normal of electrode surface) and

water normal orientation (θ_{normal} , the angle formed between the normal of water plane and the normal of electrode surface) are evaluated to characterize the local structure of interfacial water molecules. In general, the interfacial water orientation tends to be vertical at the negatively polarized electrode, but water molecules lie flat on the positively polarized electrode. At negatively charged electrode, the study reveals some difference in water orientations for different RTILs, but no such differences were clearly observed for the positively charged electrode.

Specifically, at negatively charged gold electrode, the dipole orientation (highest probability at 152°) of interfacial water molecules in hydrophobic [pyr14][TFSI] as well as their normal orientation (peak located at 90°) shows that most water tends to be vertical to the electrode with its two hydrogen atoms both approaching to the surface. Although water in hydrophilic [BMIM][BF₄] is also vertical to the electrode, however, it is more sideling than in [pyr14][TFSI] as one of its hydrogen atom is closer to the electrode surface than the other (dipole orientation shows two probability peak located at 90° and 154° , respectively). Near positively charged electrodes, for both RTILs and electrodes, the normal orientations (which means the distribution of angle formed between the normal of water plane and the normal of electrode surface), showing two peaks located at 16° and 164° as well as the dipole orientations with peak location at around 75° , depict that most water adopts a configuration parallel to the electrode. It also implies that both electrode type and the hydrophobic/hydrophilic nature of RTILs have weak influence on the water orientation at positively charged electrodes.

As a result, changing electrode type from gold to carbon does not alter much about the trend of these observations, which further suggests the minor role of electrode type in water electrosorption.

Note that a vertical configuration of water indicates that the dipole of water tends to align parallel to the electrical field created by charged electrode, which gives a lower potential energy between water and electrified electrode of any sign. The water orientation analysis here is actually in line with our component-analysis of interaction energy performed in main text, in which the coulombic interaction energy of water with negative electrode is always lower than that of water with positive electrode when under the very close absolute value of the potential drop across the EDL (this comparison is eye-guided by black arrows in the following **Figure R1.2**).

Figure R1.2 (copied from Figure 5 in main text) | The weight of different components of the total interaction energy of water molecule adsorbed at the electrode surface under different electrode polarizations. a-d, The figures for water adsorbed from [pyr14][TFSI] (a) and [BMIM][BF₄] (b) near gold electrode and from [pyr14][TFSI] (c) and [BMIM][BF₄] (d) near carbon electrode. Red, yellow, blue, and green bar charts represent water-ions van der Waals, respectively. The magenta dots connected by solid lines are the total value of “local” potential energy of water with its nearby species in the interfacial region, which results from the summation of previous four interactions.

To delve deeper into the association between interfacial water and nearby RTIL ions, we analyze the number of hydrogen bonds (H-bonds) that one interfacial water molecule could form with surrounding cations/anions with respect to EDL potential on both gold and carbon electrode (**Figure R1.3**). Herein, for a cation we define an H-bond between a hydrogen atom of cation and an oxygen atom of water to be formed if the distance between them, R_c , is shorter than 0.35 nm and the angle of hydrogen–carbon–oxygen, θ_c , (each carbon atom in cation could be donor and oxygen atom in water is acceptor) is less than 30 degrees¹⁰. For an anion, an H-bond is

determined between a hydrogen of water and an electronegative atom of anion (acting as acceptor, here are fluorine, nitrogen or oxygen atom).

As shown in **Figure R1.3**, firstly, one can observe that water has stronger hydrogen bonding with [BMIM][BF₄] than [pyr14][TFSI] as the number of H-bonds formed by water-BMIM⁺ and water-BF₄⁻ are larger than the number of H-bonds formed by water-pyr14⁺ and water-TFSI⁻, respectively. Secondly, under zero polarization, water forms much more H-bonds with anions than with cations, suggesting a dominant role of anion in water-RTIL interaction.

Figure R1.3 | Hydrogen bonds formed by RTIL ions and interfacial water as a function of applied potential. **a-b**, Number of H-bond by cations/anions in [pyr14][TFSI] (**a**) and in [BMIM][BF₄] (**b**) with interfacial water molecules at gold electrodes. **c-d**, Number of H-bond by cations/anions in [pyr14][TFSI] (**c**) and in [BMIM][BF₄] (**d**) with interfacial water molecules at carbon electrodes. The images above panel (**a**) and (**b**) show the schematics of H-bond by water molecules with cations and anions in [pyr14][TFSI] and [BMIM][BF₄], respectively. Blue dashed lines are for H-bond of water-cation, and red solid lines represent for H-bond of water-anion.

When the electrode surface becomes more negatively electrified, for hydrophobic [pyr14][TFSI] the number of water-pyr14⁺ H-bonds remains a very small value (**Figure R1.3a**),

and that of water–TFSI⁻ H-bonds decreases dramatically (e.g., from 1.12 to 0.10 on gold electrode, **Figure R1.3c**). The scenario of [BMIM][BF₄] is much different (**Figure R1.3b,d**): water and BF₄⁻ still form more H-bonds than water with BMIM⁺ even in BMIM⁺-rich double layers (for instance, 0.8 vs. 0.46 on gold electrode). It is worthy noting that the double layer formed near negative electrode features an alternative layering of ions, in which the 1st layer is cation and 2nd layer is anion. The non-increasing number of water–BMIM⁺ H-bonds indicates that water weakly associate with BMIM⁺; while the considerable margin of water–BF₄⁻ H-bonds in cation-rich double layer reveals that water is strongly attracted by anions BF₄⁻, which makes the adsorption of water on negatively charged electrode unstable. This can help explain the left peak shifting in dipolare orientation of water absorbed in [BMIM][BF₄] as compared to water incorporated in [pyr14][TFSI], because one O-H bond in water would point to BF₄⁻ layer to form H-bonds (see red solid lines in the left panels of previous **Figure R1.1a** and **R1.1c**).

When the EDL potential gets more positive, the number of water–anion H-bonds increases a little bit, while each water could form fewer hydrogen bonds with cations (for [BMIM][BF₄]) or maintain at a quite low level (for [pyr14][TFSI]). This can be ascribed to the anion accumulation and cation removal in EDLs near positively charged electrodes (see **Figure 1** of the main text and **Supplementary Figure 4**). Illustrated by the normal orientation of the interfacial water under positive polarization shown in the **Supplementary Figure 12**, water tends to lie on the electrode surface, as water forms strong hydrogen bonding with the surrounding anions.

Moreover, either cation or anion in hydrophilic [BMIM][BF₄] could have stronger hydrogen bonding with water than in hydrophobic [pyr14][TFSI], over the whole potential range. This observation is in line with previous studies reporting that water molecules would be associated with cations, however, they have stronger interaction with anions, which also account for why anions dominate the miscibility of RTILs.^{11,12}

The electrode type seems to have minor influence on both orientation of interfacial water and the hydrogen bonding between interfacial water and nearby cations/anions. This can help explaining why gold and carbon electrode have similar features of water electrosorption in MD simulations.

We find that the structure of water located in the interfacial region is very interesting and worth of deep investigation. Based on the reviewer's kind advice, both orientation and hydrogen

bonding analysis of interfacial water have been carried out and integrated into revised manuscript and **Supplementary Information**.

Brief summary:

Discussions on orientation and H-bond of water have been integrated into the revised manuscript (*Results Section of main text, page 11-12, lines 259-283*), as:

“As PMFs demonstrate where water would be predominantly distributed, we then focus on the structures of water within the interfacial region. Since water in hydrophobic and hydrophilic RTILs undergoes different association with cations/anions, one can expect that the structures of interfacial water may depend on the nature of the RTILs, EDL potential and electrode type. Revealed by the orientations of water in [pyr14][TFSI] and [BMIM][BF₄] at the electrified electrode (Supplementary Fig. 12), it could be concluded that 1) the interfacial water tends to orient vertically to the negative electrode but lies flat on the positive electrode; 2) for negatively charged electrode water orientation in different RTILs exhibits some differences, but no notable differences are seen at the positively charged electrode. Since the surface charges would always force the dipole of water to align vertically to electrode surface, the discrepancy of water orientation at cathode and anode would be attributed to the different water-RTIL associations, to a large extent, determined by water-ion hydrogen bonding. Detailed analysis of hydrogen bonding between water and cations/anions in the interfacial region, which can be found in Part G of Supplementary Information (Supplementary Fig. 13), unveils that 1) water forms more H-bonds with anions than cations, even in the cation-rich EDL under negative polarization, suggesting that anions play a dominant role in hydrophobicity/hydrophilicity of RTILs; 2) more water-ion H-bonds are observed in hydrophilic [BMIM][BF₄] than in hydrophobic [pyr14][TFSI], consistent with previous work.^{45,46}

The overall water-RTIL hydrogen bonding as well as water orientation strongly depends on the potential drop across the EDL. Thus, naturally, the electrosorption of water depends not only on the water-ion interaction, ‘water following anions’, but also on the water-electrode interaction.”

New references on water-RTILs interaction are added, as:

“45. Cammarata, L., Kazarian, S. G., Salter, P. A. & Welton, T. Molecular states of water in room temperature ionic liquids. *Phys. Chem. Chem. Phys.* **3**, 5192–5200 (2001).

46. Anthony, J. L., Maginn, E. J. & Brennecke, J. F. Solution thermodynamics of imidazolium-based ionic liquids and water. *J. Phys. Chem. B* **105**, 10942–10949 (2001).”

More detailed descriptions are provided in the updated **Supplementary Information** (*Supplementary Part G. Orientation and H-bond of water adsorbed at the electrode*).

Comment 3) Even if there is only one graphene layer in their electrode, I am not sure that real "graphene" would behave as shown in this study. Due to the accessibility of the two faces of the plane, peculiar adsorption behaviors have recently been shown in molecular dynamics studies (both for water and for ionic liquids). I think the material simulated by the authors is somewhat more similar to graphite, so that I recommend to switch graphene to graphite in the figures and in the text.

Response:

Thank the reviewer for pointing out this inaccuracy in terminology (a slang often used by ‘simulators’. We have now systematically replaced ‘graphene’ term with ‘graphite’.

Comment 4) Why didn't the authors perform similar experiments on a carbon electrode? Here they only did gold, but being able to do the same comparison as in the simulations would strengthen the conclusions.

Response:

We thank the reviewer for this valuable advice. Following this advice, the effect of water sorption on electrochemical activity at carbon electrode was investigated by using **Highly Oriented Pyrolytic Graphite (HOPG)** as working electrode. The surface structure of HOPG is very close to the one used in the simulation model. A Pt wire and an Ag wire serve as counter electrode and reference electrode, respectively.

CV curves of HOPG in hydrophobic [pyr14][TFSI] were shown in **Figure R1.4**. It can be seen that when water content increased from 570 to 3815 ppm, reduction current abruptly increased at around -0.7 V, which was not observed in [pyr14][TFSI] containing 570 ppm water, so that the electrochemical window decreases a lot. Since all experimental conditions except water content was kept unchanged, the decrease of electrochemical window with increasing water content should be ascribed to the reduction of water. However, for hydrophilic [BMIM][BF₄], every similar CV curves were obtained when water content increased from 590 to 3915 ppm (see **Figure R1.4b**). No significant reduction current was observed and the electrochemical window is nearly the same under different water content. The above results

demonstrate that carbon electrode presents the same trend as Au electrode concerning the effect of water sorption on electrochemical activity in RTILs.

Figure R1.4 | The effect of water sorption on electrochemical activity on carbon electrode. a-b, Cyclic voltammograms of HOPG in hydrophobic [pyr14][TFSI] (a) and hydrophilic [BMIM][BF₄] (b) under different water contents. Scan rate: 100 mV/s.

Brief summary:

The original Abstract has been slightly modified as we added experiments on HOPG (*page 1, lines 17-18*), as:

“This conclusion is verified by electrochemical cyclic voltammetry measurements on gold and carbon electrodes”

Discussions related to CV measurements on HOPG have been integrated into the revised manuscript (*Result Section of main text, page 9, lines 217-221*), as:

“CV measurements on highly oriented pyrolytic graphite (HOPG) electrodes were also investigated to explore the impact of electrode type, which exhibit the same trend as seen for Au(111) electrode concerning the effect of water electro-sorption on electrochemical activity in hydrophobic and hydrophilic RTILs (Supplementary Fig. 8).”

The description of the process of obtaining the HOPG electrode has been added into the manuscript (*Method Section of the main text, page 20, lines 487-489*):

“HOPG electrode was purchased from Bruker. A piece of tape was pressed onto the flat surface and then pulled it off. The freshly cleaved surface of HOPG was used for measurements.”

Detailed experimental result and descriptions of CV measurements on HOPG can be found in the update **Supplementary Information** (*Supplementary Part D. Effects of the water concentration and electrode material*).

Comment 5) I do not agree with the statement line 214. The PMFs shown on Figure 4 do not correspond to a simple "potential energy". This should be corrected.

Response:

We thank the reviewer for this suggestion. We changed ‘potential energy’ as ‘mean force potential’ (*page 10, lines 223*), as:

“we evaluated the **mean force potential** of a water molecule as a function of the distance to the electrode surface under different EDL potentials.”

Comment 6) Minor point: Line 379 please correct "inasmuch"

Response:

We thank the reviewer for pointing this out. We have corrected it.

Responses to Reviewer #2's Comments

Reviewer #2 (Remarks to the Author):

The manuscript “Reducing water electrosorption to electrodes from humid ionic liquids” focusses on the electrosorption behavior of water in ionic liquids on two different surfaces (gold and graphene). The authors use two different ionic liquids (one hydrophilic and one hydrophobic) and study water adsorption behavior by complementary molecular dynamic (MD) simulation and cyclic voltammetry (CV). The authors conclude that water will be depleted on negatively charged electrodes in a hydrophilic ionic liquid system in general.

While the presented work appears to be very well done, I feel that the authors do not have enough data and control experiments to support a general conclusion that warrants publication in a journal such as Nat. Comm. Currently, the data is good enough for a more specialized outlet after major revisions. Hence, I feel that the work is currently premature for consideration in Nat. Comm. and generally not ready for acceptance. However, considering the potential interest in ionic liquids (and the effect of water in ILs), a substantially revised version with appropriate control experiments may be reconsidered for publication in Nat. Comm. My major concerns are as follows:

Response:

We thank the reviewer a lot for carefully assessing our work and giving quite constructive suggestions that more data and control experiments are needed to support our conclusions.

Comment 1) My first doubt about generalization is the specific water level used: Is there any particular reason for using about 3000 ppm of water in both ILs? Is this possibly the maximum which the hydrophobic IL can absorb? I can see that one wants to compare similar levels. Yet, what would be the effect of a higher concentration (if possible). Or more general: Is there a concentration dependence of the observed effect?

Response:

We appreciate the reviewer for those valuable comments on the effects of water concentration.

For reviewer's first two questions, it is true that the water content in hydrophobic IL has upper limit of maximum water concentration. To better illustrate the difference of [pyr14][TFSI] and [BMIM][BF₄] in their hygroscopicity, we carried out experiments in which the change of water content of initially dried RTILs with time is monitored (see the following **Figure R2.1**). It can be seen that the water concentration that [pyr14][TFSI] can reach under normal environment after two days is approximately 3500 ppm, while the value of [BMIM][BF₄] is ~30000 ppm. In addition, we also measured water concentration in ILs after an exposure to air for one week, and

found water concentrations are about 3900 and 33000 ppm for [pyr14][TFSI] and [Bmim][BF₄], respectively (the data were not shown on the curves).

Figure R2.1 | Hygroscopicity of RTILs in humid air. a-b, Water content in [pyr14][TFSI] (a) and [BMIM][BF₄] (b) as a function of time. Red solid lines represent the averaged humidity during the measurements; Blue lines are to guide the eyes.

As also pointed by the reviewer, it would be interesting and necessary to investigate the water concentration effects on water electro sorption, especially to see if the conclusions still hold under very high concentration. Hence, to be on the safe side, we systematically perform MD simulations and carry out more cyclic voltammograms (CV) measurements for these two RTILs with different water concentrations (based on the above results of RTIL hygroscopicity) to investigate the water-concentration dependence.

In MD simulations (see the following **Figure R2.2**), the water concentration effects on the dependence of water electro sorption vs electrode potential, are small for these two studied RTILs on both gold and carbon electrodes. For hydrophobic RTIL [pyr14][TFSI] (**Figure R2.2a,c**), increasing water concentration from 2796 ppm to 3910 ppm does not change the water electro sorption curves much, regardless of electrode materials. For the hydrophilic [BMIM][BF₄] (**Figure R2.2b,d**), either increasing or decreasing the water concentration does not alter the trends in water electro sorption. Even if the water content reaches a very high level (22838 ppm), depletion of water could still be observed at the negative polarization.

Figure R2.2 | Electrodesorption of water from RTILs with different water contents on gold and carbon electrodes. Panels (a) and (b) show water adsorption from humid [pyr14][TFSI] and [BMIM][BF₄] with different water content on gold electrodes, respectively; Panels (c) and (d) are figures for these two humid RTILs on carbon electrodes.

Correspondingly, the concentration-dependent experiments were carried out as well. In particular, based on the reviewer's suggestion of "control experiment", RTILs with different water content could have either the same cation or the same anion for comparison (two new RTILs were chosen as [BMIM][TFSI] and [BMIM][OTf]).

Compared with the data in the previous version of our manuscript (only dry and humid RTILs were tested), data for RTILs [pyr14][TFSI] and [BMIM][BF₄] with three different water concentrations are presented now (see **Figure R2.3**), which benefit understanding the effect of water electroadsorption on electrochemical windows of humid RTILs. Therefore, CV curves in **Figure 3** of the previous manuscript were updated. It should be pointed out that the additional experiments adopted newly purchased [pyr14][TFSI] (IoLiTec) in the highest available quality (99%), performed, as previously, in the same processing way and using the same measurement

method, but the lower and upper limits of electrochemical windows were better controlled. From the additional CV measurements, we got a reliable, reproducible set of data, which are presented in the revised manuscript.

Specifically, from the updated CV measurements for [pyr14][TFSI] and [BMIM][BF₄], it can be seen that: 1) the electrochemical window of hydrophobic [pyr14][TFSI] significantly reduces as water concentration increases from 538 to 3815 ppm (**Figure R2.3a**). 2) For hydrophilic [BMIM][BF₄], however, CV curves exhibit similar characteristics and nearly the same electrochemical windows are obtained over a wide range of water concentration from 14 to 22800 ppm (**Figure R2.3b**). Especially, water concentration in [pyr14][TFSI] is quite lower than the large value used for the test in [Bmim][BF₄] (3815 vs 22800 ppm). In addition, CV measurements for these two RTILs on highly oriented pyrolytic graphite (HOPG) electrodes were also performed, which exhibit the same trend as Au(111) electrode concerning the effect of water electrosorption on electrochemical windows of humid RTILs (see **Supplementary Fig. 8**).

Figure R2.3 | The effect of water sorption on electrochemical activity on gold electrode. a-b, Cyclic voltammograms of Au(111) in hydrophobic [pyr14][TFSI] (a) and hydrophilic [BMIM][BF₄] (b) under different water contents. Scan rate: 100 mV/s.

Furthermore, we examined the scenarios of CV measurements on another two RTILs – [BMIM][TFSI] and [BMIM][OTf] (see the following **Figure R2.4**). Please note that the RTIL [BMIM][TFSI] was also newly purchased from IoLiTec in the highest available quality (99%), and [BMIM][OTf] was newly purchased from Lanzhou Yulu Fine Chemical Co., LTD (in 99% quality). RTIL [BMIM][TFSI] is hydrophobic while RTIL [BMIM][OTf] is hydrophilic.^{13–15}

For hydrophobic RTIL [BMIM][TFSI], the electrochemical behavior that the increase of water content obviously decreases the electrochemical window is consistent with that of hydrophobic RTIL [pyr14][TFSI] (compare **Figure R2.4a** with **Figure R2.3a**). As expected, for hydrophilic RTIL [BMIM][OTf], similar cyclic voltammograms were obtained when water content is increased from 1230 to 3966 ppm, which is consistent with the trend in hydrophilic RTIL [BMIM][BF₄] on Au(111) electrode (compare **Figure R2.4b** with **Figure R2.3b**).

Figure R2.4 | The effect of water sorption on electrochemical activity on gold electrodes. a-b, Cyclic voltammograms of Au(111) in hydrophobic [BMIM][TFSI] (a) and hydrophilic [BMIM][OTf] (b). Scan rate: 100 mV/s.

Note that in practical applications, hydrophilic RTILs may inevitably absorb more water than hydrophobic RTILs during the synthesis, storage and implementation process. However, the above investigations on water concentration effects demonstrate that the electrochemical window of hydrophobic RTILs is reduced as the water content increases, since water molecules eagerly adsorb onto the electrified electrode surface and thus favor the electrochemical reaction of water. For hydrophilic RTILs, their electrochemical windows can still be well exploited without water reaction even if they could in nature absorb more water. These findings not only support our conclusions of the previous version of the manuscript, but also indicate that the capability of hydrophilic RTILs to maintain their electrochemical window is robust over a wide range of water concentration. This fact is nontrivial, and we consider it as one of the highlights of the paper.

Brief summary:

Discussions related to the influence of different water concentration on water electrosorption have been incorporated into the revised manuscript (*Results Section of main text, page 8, lines 181-185; page 8-9, lines 194-221; Figure 3; page 16, lines 367-375*), as:

“Additional simulations on these two studied RTILs with different water content reveal that the influence of water concentration on the observations in Fig. 2 is weak (Supplementary Fig. 7). Even under ~23000 ppm of water in humid RTIL [BMIM][BF₄], water depletion could still be observed at negatively charged electrode (Supplementary Fig. 7b).”

“For humid RTIL [pyr14][TFSI] (Fig. 3a), when water content is increased from 538 ppm to 1260 ppm, the electrochemical window is narrowed to some extent; with water content enlarged from 1260 to 3815 ppm, the decrease of electrochemical window was observed distinctly: the lower limit of electrochemical window gets moved from around -2.4 to -1.9 V. Since all experimental conditions except water content was kept unchanged, it is reasonable to attribute the decrease of electrochemical window to the reduction of water. When potential scans positively, the characteristic peaks are present. Taking 1260 ppm water content as an example, an anodic current wave at around +2.0 V and a cathodic peak at about +1.4 V were observed, which stand for the oxidation of Au(111) surface and the corresponding reduction, respectively.

Figure 3 | The effect of water electrosorption on electrochemical activity on gold electrode. a-b, Cyclic voltammograms of Au(111) in hydrophobic [pyr14][TFSI] (a) and hydrophilic [BMIM][BF₄] (b) under different water contents. Scan rate: 100 mV/s.

On the contrary, for humid RTIL [BMIM][BF₄] CV characteristics are similar as water content changes from 14 to 3014 and then to 22800 ppm. As shown in Fig. 3b, electrochemical windows remain nearly unchanged even considering the remarkable change of water content.”

“Although pyr14⁺ and BMIM⁺ are very different in terms of their molecular structure, the observations in hydrophobic [BMIM][TFSI] (Supplementary Fig. 14a and 15a), both in MD simulations and experiments, follow the same trend uncovered for hydrophobic [pyr14][TFSI]. On the other hand, similar to hydrophilic [BMIM][BF₄], one can see that water in hydrophilic [BMIM][OTf] stays away from the negative electrodes (Supplementary Fig. 14b), so that the electrochemical window could be maintained with the increase of water content (Supplementary 15b). This fact is very important to know, if one wants to use RTILs as electrolytes in supercapacitors or a solvent for electrocatalysis.”

More details on water concentration effects on simulations are provided in the updated **Supplementary Information** (*Supplementary Part D. Effects of the water concentration and electrode material; Part H. Explorations on additional RTILs*).

The hydrophilicity/hydrophobicity of RTIL [pyr14][TFSI] and [BMIM][BF₄], as well as their hygroscopicity can be found in the update **Supplementary Information** (*Supplementary Part B. Hydrophobicity/hydrophilicity and hygroscopicity of RTILs*).

Comment 2) Mentioning the maximum water content at room temperature will be interesting. In particular, considering that the authors label their ILs hydrophilic and hydrophobic. What is the criterion for this label? Do they mean water soluble vs. not water soluble? Or do they refer to the ability to take up water above a certain threshold?

Response:

We thank the reviewer for these sharp comments. Indeed, these points need clarification, whereas we did it ‘in passing’.

For the first comment, based on our water content–time experiments, the maximum water content that the dry [pyr14][TFSI] and [BMIM][BF₄] could reach after being exposed to the air for two days is ~3500 ppm and ~30000 ppm of water, respectively. It is clear that the hygroscopicity of these two RTILs differs a lot. The details could refer to **Figure R2.1** and corresponding descriptions/discussions in aforementioned response to **Comment 1**.

We shall admit that to the best of our knowledge, the clear definition of hydrophobicity/hydrophilicity of RTILs can hardly be found in existing research. However, the label “hydrophobic and hydrophilic” have been widely used in previous studies, in which they are associated with the miscible and immiscible features of RTILs.^{13,16–18} For instance, Huddleston *et al.*¹⁶ measured the water content of various RTILs that have been dried after certain procedures and stored in contact with water, respectively (see **Table R2.1** that snipped from their paper that has **been cited around 3000 times**), and pointed out that anion identity is of foremost importance for RTILs hydrophilicity.

Ionic liquid	Water equilibrated/ppm	Dried/ppm
[C ₄ mim][Cl]	Miscible	2200
[C ₄ mim][I]	Miscible	1870
[C ₄ mim][BF ₄]	Miscible	4530
[C ₄ mim][PF ₆]	11700	590
[C ₄ mim][Tf ₂ N]	3280	474
[C ₆ mim][Cl]	Miscible	1130
[C ₆ mim][PF ₆]	8837	472
[C ₈ mim][Cl]	Miscible	890
[C ₈ mim][PF ₆]	6666	388

^a Water equilibrated refers to RTILs that have been stored in contact with water. Dried RTILs are water equilibrated RTILs that have been dried at 70 °C for 4 h on a vacuum line.

Table R2.1 | Water content of several water equilibrated (25 °C) and dried ILs. This Table is snipped from paper by Huddleston, *et al.*, titled “Characterization and comparison of hydrophilic and hydrophobic room temperature ionic liquids incorporating the imidazolium cation”. *Green Chem.* 3, 156–164 (2001)”. This paper has been cited around 3000 times.

We therefore examine the miscibility of our studied [pyr14][TFSI] and [BMIM][BF₄] *via* both MD simulations and experiments (see the following **Figure R2.5**). MD simulations are all performed at 298K (Nos é-Hoover thermostat) and 1 bar (Berendsen barostat) in NPT ensemble, and the water-to-RTIL molar ratio is ~15:1. Specifically, simulation of water/[pyr14][TFSI], starting from “randomly mixed” water with [pyr14][TFSI], shows the process of demixing, in which water and RTIL are separated into two phases (see **Figure R2.5a** and **Supplementary Movie 1**). Simulation of water/[BMIM][BF₄], starting from “initially separated” water and

[BMIM][BF₄], however, shows that the two liquids could spontaneously mix with each other (**Figure R2.5c** and **Supplementary Movie 2**).

Figure R2.5 | Hydrophobicity and hydrophilicity of RTILs. a-b, MD system snapshots (a) and experimental image (b) of water and hydrophobic RTIL [pyr14][TFSI]. c-d, MD system snapshots (c) and experimental image (d) of water and hydrophilic RTIL [BMIM][BF₄].

In experiments, 2 mL water was added into 2 mL each of RTILs and the mixture was stirred. Then, the water/RTIL mixture was sealed and stored for two hours. The miscibility of [pyr14][TFSI] and [BMIM][BF₄] are determined optically. Although the two RTILs and water are colorless and transparent, the water/[pyr14][TFSI] interface can be observed clearly (**Figure R2.5b**). However, water/[BMIM][BF₄] mixture formed a homogeneous phase (**Figure R2.5d**).

In conclusion, both simulation and experimental results show that RTIL [pyr14][TFSI] is water-immiscible while RTIL [BMIM][BF₄] is miscible with water. Hence, [pyr14][TFSI] can be classified into hydrophobic RTILs while [BMIM][BF₄] is hydrophilic.

Brief summary:

Discussions on hydrophobic/hydrophilic nature of RTILs have been modified in the revised manuscript (*Page 2, lines 41-46; Page 4, lines 100-103*), as:

“Many RTILs are water-miscible, labeled conventionally as hydrophilic, for instance, the ‘popular’ 1-Butyl-3-methylimidazolium tetrafluoroborate ([BMIM][BF₄]).^{12,13} But even some RTILs that are ‘more hydrophobic than others’, such as RTIL 1-Butyl-1-methylpyrrolidinium bis(trifluoromethylsulfonyl)imide ([pyr14][TFSI]) which is immiscible with water, are still hygroscopic: they absorb water up to their water concentration saturation level.¹²⁻¹⁴”

“The hydrophobicity and hydrophilicity as well as hygroscopicity of RTILs were examined by both MD simulation and experiment of water-RTIL mixture (Supplementary Fig. 2 and Fig. 3), showing that the hydrophobic RTIL [pyr14][TFSI] is immiscible with water while the hydrophilic RTIL [BMIM][BF₄] is water-miscible.”

New references to the hydrophobicity/hydrophilicity of RTILs have been added, as:

“13. Huddleston, J. G. *et al.* Characterization and comparison of hydrophilic and hydrophobic room temperature ionic liquids incorporating the imidazolium cation. *Green Chem.* **3**, 156–164 (2001).

14. Appetecchi, G. B., Scaccia, S., Tizzani, C., Alessandrini, F. & Passerini, S. Synthesis of Hydrophobic Ionic Liquids for Electrochemical Applications. *J. Electrochem. Soc.* **153**, A1685 (2006).”

Descriptions that are more detailed can be found in the updated **Supplementary Information**, in which we provide: 1) details of related MD simulations and experiments. 2) Characterization of the miscibility of the two RTILs, and 3) their maximum water content after a two-day exposure in room environment (*Supplementary Part B. Hydrophobicity/hydrophilicity and hygroscopicity of RTILs; Supplementary Movie 1 and Movie 2*).

Comment 3) The authors state that the hydrophilicity of the ionic liquid is responsible for the observed behavior. However, the so-called “hydrophobic/hydrophilic” character of the IL itself (however the authors wish to define this) may not be the correct descriptor for the observed behavior. I am not yet convinced, and as the devil’s advocate I argue that the actual molecular structure of the cations plays a more central role in this case. Specifically, BMIm⁺ has planar delocalized charge (pi orbital delocalization) and pyr14⁺ has a point charge on a tertiary ammonium, which certainly leads to different interaction/packing behavior at the surface and to different interactions with water itself. As such, it seems equally likely that the observed effects are cation specific, rather than being connected to the hydrophilicity of the liquid itself.

Response:

We thank the reviewer for these helpful comments. It is true that in our previous manuscript, we have not shown or discussed the influence from the cation type. Therefore, we carried out MD

simulations and CV measurements on another RTIL with a different cation but the same anion (i.e., [BMIM][TFSI] was selected to compare with [pyr14][TFSI]). The new results are capable to deliver a demonstration that the observed phenomenon of interfacial water electroadsorption is not cation/anion specific, and mainly depends on the hydrophilicity of RTILs.

As already discussed in above response to **Comment 2**, hydrophilic RTILs are miscible with water while the water concentration in hydrophobic RTIL has an upper limit. The hydrophilicity (or miscibility) of RTIL is mainly determined by the anion; the cation can affect the hydrophobicity (immiscibility) but only change the maximum water content rather than switch between hydrophobicity and hydrophobicity.^{13,16} For example, for PF_6^- -based imidazolium RTILs, increasing the alkyl chain length from butyl to hexyl to octyl increases the hydrophobicity (see the **Table R2.1** shown in previous response);¹⁶ with the same anion, RTILs with pyridinium-based cation are more hydrophobic than RTILs with imidazolium-based cation.¹³ To know the specific hydrophobicity of RTIL used in this work, we have performed MD simulations and CV experiments of RTIL [BMIM][TFSI], as it differs in cation type but has the same anion with RTIL [pyr14][TFSI], so that the role of cation in water electroadsorption can be properly assessed.

Figure R2.6 | Electroadsorption of water in humid RTILs on gold electrodes, revealed by simulation. Panels (a) and (b) show water adsorption on Au(111) from humid [pyr14][TFSI] and [BMIM][TFSI], respectively. The relative enrichment or depletion of water molecules in the first interfacial layer is depicted by the ratio of averaged number density of water in this layer to that of water in the bulk region. Enrichment (depletion) zone corresponds to a higher (lower) water density than in the bulk. Water content in [pyr14][TFSI] and [BMIM][BF₄] is 2796 ppm and 2865 ppm, respectively.

Revealed by MD simulations, water electroadsorption with respect to the EDL potential for [BMIM][TFSI] and [pyr14][TFSI] (both of them are hydrophobic) follow the same trend (see

Figure R2.6). The enrichment of water electro sorption in [BMIM][TFSI] occurs as the electrode potential rises, which is akin to the observations for [pyr14][TFSI]. Similarly, in experiments, CV curves for [BMIM][TFSI] and [pyr14][TFSI] both demonstrate that increasing water concentration would obviously reduce their electrochemical window (see **Figure R2.7**).

Figure R2.7 | The effect of water sorption on electrochemical activity on gold electrode revealed by experiment. a-b, Cyclic voltammograms of Au(111) in hydrophobic [pyr14][TFSI] (a) and [BMIM][TFSI] (b). Scan rate: 100 mV/s.

In conclusion, previous work revealed that cation does affect the hydrophobicity, but the miscibility of RTIL (in other word, hydrophilicity) is mainly dominated by anion type.^{13,16} Our MD simulations show that two hydrophobic RTILs, with different cation type (imidazolium and pyrrolidinium-based) but the same anion (TFSI), follow similar water electro sorption pattern. Experiments further support that in both cases, even a very small amount of water could lead to noticeable reduction in their electrochemical windows. These results give strong evidence that the observed phenomenon is determined by the hydrophilicity of RTILs rather than being cation specific.

Brief summary:

Discussions on MD simulations and CV measurements of RTIL [BMIM][TFSI] have been added into the revise manuscript (*Results Section* of main text, page 16, lines 367-375), as:

“Although pyr14⁺ and BMIM⁺ are very different in terms of their molecular structure, the observations in hydrophobic [BMIM][TFSI] (Supplementary Fig. 14a and 15a), both in MD simulations and experiments, follow the same trend uncovered for hydrophobic [pyr14][TFSI]. On the other hand, similar to hydrophilic [BMIM][BF₄], one can see that water in hydrophilic

[BMIM][OTf] stays away from the negative electrodes (Supplementary Fig. 14b), so that the electrochemical window could be maintained with the increase of water content (Supplementary 15b). This fact is very important to know, if one wants to use RTILs as electrolytes in supercapacitors or a solvent for electrocatalysis.”

Details on related MD simulations and CV curves can be found in the updated **Supplementary Information** (*Supplementary Part H. Explorations on additional RTILs*).

Comment 4) In the same direction, the role of the anion is not discussed at all, even though the anions are in this case partly responsible for the overall hydrophobic/hydrophilic character of the liquids. The authors compare two ILs with two cations AND two different anions (a hydrophilic and a very hydrophobic anion). Hence, the overall character of the liquids is equally affected by the anions. As such, making a generalizing statement utilizing the overall liquid character seems very weakly supported by evidence at this point.

Response:

We thank the reviewer for this comment because it has stimulated us to deepen the analysis of this question, although we addressed some discussions on the role of anions in the previous version of our manuscript. To be specific, as suggested by the reviewer, "control experiment" on RTILs (those have the same cation but different anion and the same anion but different cation) was carried out, to better unveil anion effects on water electrosorption.

Following the discussion in above response to **Comment 3**, besides [BMIM][TFSI], we performed MD simulations and CV measurements for another hydrophilic [BMIM][OTf] to further control the anion's impact. The reason why we choose for this the hydrophilic [BMIM][OTf] rather than hydrophilic [BMIM][Cl] suggested by the reviewer in **Comment 5** is that the melting point of RTILs with chloride elements is typically over the room temperature,¹⁹ whereas RTIL [BMIM][OTf] has low melting point and it can easily be simulated at room temperature. In addition, their electrochemical window of chloride containing RTILs is relatively narrow compared with the frequently-used RTILs such as [pyr14][TFSI] and [BMIM][BF₄], due to the occurrence of chlorine evolution reaction.²⁰

The overall comparison of results for [BMIM][TFSI], [BMIM][BF₄] and [BMIM][OTf] can fairly reveal the role of anion.

Figure R2.8 | Electrosorption of water in humid RTILs on gold electrodes, revealed by simulation. Panels (a), (b) and (c) show water adsorption on Au(111) from humid [BMIM][TFSI], [BMIM][BF₄] and [BMIM][OTf], respectively. The relative enrichment or depletion of water molecules in the first interfacial layer is depicted by the ratio of averaged number density of water in this layer to that of water in the bulk region. Enrichment (depletion) zone corresponds to a higher (lower) water density than in the bulk.

In MD simulations, water molecules hydrophobic RTIL [BMIM][TFSI] would adsorb on the electrified electrode of any sign, as the absolute value of EDL potential increases (see **Figure R2.8a**). However, both hydrophilic RTILs ([BMIM][BF₄] and [BMIM][OTf]) feature the depletion of water electrosorption under negative polarization, suggesting that water prefers to stay in bulk region instead of being adsorbed on the negative electrode (see **Figure R2.8b** and **R2.8c**). In CV measurement of hydrophobic [BMIM][TFSI], it can be observed that the electrochemical window can be apparently reduced with higher water concentration (from 410 to 3090 ppm of water, see **Figure R2.9a**). Nevertheless, the scenarios of hydrophilic [BMIM][BF₄] and [BMIM][OTf] are quite different, as their electrochemical window are not affected by the increasing water content (from 14 to 22800 ppm of water, see **Figure R2.9b**; from 1230 to 3966 ppm of water, see **Figure R2.9c**).

Figure R2.9 | The effect of water sorption on electrochemical activity on gold electrode revealed by experiment. a-b, Cyclic voltammograms of Au(111) in hydrophobic [BMIM][TFSI] (a) and hydrophilic [BMIM][BF₄] (b) and hydrophilic [BMIM][OTf]. Scan rate: 100 mV/s.

In conclusion, we carried out explorations on another two RTILs: the hydrophobic [BMIM][TFSI] and hydrophilic [BMIM][OTf]. The simulation-obtained results and experimental data, compared with the updated data of hydrophobic [pyr14][TFSI] and hydrophilic [BMIM][BF₄] that have been integrated into the revised manuscript, have confirmed the conclusion that hydrophilic rather than hydrophobic RTILs would help to keep water molecules away from the negatively charged electrodes. In particular, these two studied hydrophilic RTILs [BMIM][BF₄] and [BMIM][OTf] consist of anions distinctive in their molecular structure, suggesting that the depletion of water at negative electrode is not principally anion-specific. Based on the explorations of all four RTILs, it seems that the phenomenon that the electrochemical window of hydrophilic RTILs is ‘waterproof’ would be general. Of course, more testifications with different RTILs would make this conclusion universal, but we hope that our work will trigger a flow of works in this direction.

Since in the following comments (in **Comment 6**), reviewer also raises the question about the association/interaction between water and surrounding cations/anions, the discussion about the role of anion in the interfacial water will be provided in the responses to **Comment 6** (the detailed molecular insights about water–cation/anion association near the electrode).

Brief summary:

Discussions related to the RTIL [BMIM][OTf] have been merged into the revised manuscript (*Results Section of main text, page 16, lines 370-375*), as:

“On the other hand, similar to hydrophilic [BMIM][BF₄], one can see that water in hydrophilic [BMIM][OTf] stays away from the negative electrodes (Supplementary Fig. 14b), so that the electrochemical window could be maintained with the increase of water content (Supplementary 15b). This fact is very important to know, if one wants to use RTILs as electrolytes in supercapacitors or a solvent for electrocatalysis.”

Detailed MD simulations and CV measurements are provided in the updated **Supplementary Information** (*Supplementary Part H. Explorations on additional RTILs*).

Comment 5) Hence, I feel that the authors must provide proper control experiments with both cations and TFSI and/or BF₄ as a common anion. As additional control, the authors will be able to further generalize by using e.g. a similar cation with TFSI and e.g. the very hydrophilic Cl⁻ anion, allowing to relate the behavior more convincingly to an overall solution behavior.

Response:

We thank the reviewer for this useful suggestion. The conclusions of the previous version of the manuscript have now been strengthened by the suggested control experiments of RTILs [pyr14][TFSI], [BMIM][BF₄], [BMIM][TFSI], and [BMIM][OTf], as well as the corresponding MD simulations, as shown in above responses to **Comments 3 & 4** and in the revised manuscript and **Supplementary Information**.

Figure R2.10 | Electrodesorption of water from humid RTILs on gold electrodes. Panels (a) and (b) show water adsorption on Au(111) from humid [BMIM][TFSI] and [BMIM][OTf], respectively. The relative enrichment or depletion of water molecules in the first interfacial layer is depicted by the ratio of averaged number density of water in this layer to that of water in the bulk region. Enrichment (depletion) zone corresponds to a higher (lower) water density than in the bulk.

Specifically, MD-revealed water electrodesorption behaviors (see **Figure R2.10**) as well as CV measurements (see **Figure R2.11**) on the two new RTILs – hydrophobic [BMIM][TFSI] and hydrophilic [BMIM][OTf], evidence, to some extent, the generality of our conclusion that using more hydrophilic RTILs could actually keep water away from the negative electrode, while water in hydrophobic RTILs willingly adsorb on the charged electrode and thus leads to reduction of electrochemical window.

Figure R2.11 | The effect of water sorption on electrochemical activity on gold electrodes. a-b, Cyclic voltammograms of Au(111) in hydrophobic [BMIM][TFSI] (a) and hydrophilic [BMIM][OTf] (b). Scan rate: 100 mV/s.

As discussed before in this response-document, RTILs [pyr14][TFSI] and [BMIM][TFSI] are both hydrophobic with different cations; RTILs [BMIM][BF₄] and [BMIM][OTf] are both

hydrophilic but different anions. The hydrophobic [BMIM][TFSI], and the hydrophilic [BMIM][BF₄] and [BMIM][OTf] can allow us to fairly assess the role of anion, as they cover the hydrophobic anion TFSI and two hydrophilic anions BF₄⁻ and OTf.

Here we did not choose RTIL [BMIM][Cl] suggested by the reviewer, mainly because [BMIM][Cl] is not in liquid phase under room temperature (the melting point of [BMIM][Cl] is over 65 °C),¹⁹ and its electrochemical window is relatively small compared with other frequently-used RTILs.²⁰

Brief summary:

Discussions related to the RTIL [BMIM][OTf] have been imbedded into the revised manuscript (*Results Section of main text, page 15-16, lines 362-382*), as:

“Generality based on different RTILs. How general is the conclusion that the electrochemical window of hydrophilic RTILs is ‘waterproof’? To answer this, MD simulations and CV measurements are both carried out for another two RTILs: [BMIM][TFSI] and BMIM-trifluoromethanesulfonate ([BMIM][OTf]); the former is hydrophobic and the latter is hydrophilic.⁴⁸⁻⁵⁰

Although pyr14⁺ and BMIM⁺ are very different in terms of their molecular structure, the observations in hydrophobic [BMIM][TFSI] (Supplementary Fig. 14a and 15a), both in MD simulations and experiments, follow the same trend uncovered for hydrophobic [pyr14][TFSI]. On the other hand, similar to hydrophilic [BMIM][BF₄], one can see that water in hydrophilic [BMIM][OTf] stays away from the negative electrodes (Supplementary Fig. 14b), so that the electrochemical window could be maintained with the increase of water content (Supplementary 15b). This fact is very important to know, if one wants to use RTILs as electrolytes in supercapacitors or a solvent for electrocatalysis.

The comparison of three RTILs studied – [pyr14][TFSI], [BMIM][TFSI], and [BMIM][BF₄], in terms of the interfacial water orientation and hydrogen bonding with RTIL ions reveals that it is anion type (which dominates the hydrophilicity of RTIL) rather than cation type that majorly determines the enrichment or depletion of water electrosorption at the negative electrode (Supplementary Fig. 16). Indeed, more RTILs should be tested to prove the universality of this effect, but its physical nature seems to be very sound.”

New references to the hydrophobicity/hydrophilicity of RTILs [BMIM][TFSI] and [BMIM][OTf] have been added, as:

“48. Cao, Y., Chen, Y., Sun, X., Zhang, Z. & Mu, T. Water sorption in ionic liquids: kinetics, mechanisms and hydrophilicity. *Phys. Chem. Chem. Phys.* **14**, 12252–12262 (2012).

49. Baj, S., Chrobok, A. & Słupska, R. The Baeyer–Villiger oxidation of ketones with bis(trimethylsilyl) peroxide in the presence of ionic liquids as the solvent and catalyst. *Green Chem.* **11**, 279–282 (2009).

50. Malik, R. S., Tripathi, S. N., Gupta, D. & Choudhary, V. Novel anhydrous composite membranes based on sulfonated poly (ether ketone) and aprotic ionic liquids for high temperature polymer electrolyte membranes for fuel cell applications. *Int. J. Hydrogen Energy* **39**, 12826–12834 (2014).”

Results, comparisons, and discussions for these four RTILs have been revised and incorporated into our revised manuscript and the **Supplementary Information**.

Comment 6) Also, in figure 1 and supplementary figure 2, the authors show MD derived polarization dependent ion distributions at electrode/IL interfaces. The correlation/interaction between water molecules and ions is not well discussed at all but also available from these simulations. Again, the structure of the ions is very different, and water interactions will certainly be different for different ions. How this correlates with the observations is unclear. I can imagine that some of the water is bound strongly to some of the anions and/or cations, which in turn of course changes the amount of overall available water for interfacial adsorption, or this may even result in co-adsorption. Again, this suggests that one has to rule out or at least understand potential water concentration (and threshold) effects, in order to support a generalizing conclusion (see again bullet point 1).

Response:

We thank the reviewer for this considerate suggestion. We have added new data and corresponding discussions related to the correlation/interaction between absorbed water and ions. Specifically, the orientations of water as well as the hydrogen bonds each water could form with surrounding cations/anions in the interfacial region have been analyzed for hydrophobic [pyr14][TFSI], [BMIM][TFSI] and hydrophilic [BMIM][BF₄].

Figure R2.11a-c display the overall comparison between the orientations of water molecules located at interfacial region from humid RTILs [pyr14][TFSI], [BMIM][TFSI] and [BMIM][BF₄] on the same type of electrode, respectively. For orientation analyses, two types of angles – water dipole orientation (θ_{dipole} , the angle formed by the water dipole vector and the normal of electrode surface) and water normal orientation (θ_{normal} , the angle formed between the normal of water plane and the normal of electrode surface) are evaluated, as depicted in **Figure R2.11f**.

Under negative polarization, hydrophobic [BMIM][TFSI] has a similar scheme of interfacial water orientation to that of hydrophobic [pyr14][TFSI], with a probability peak shift of the dipole orientation (from 152° to 130°). This suggests that cation does affect the structure of interfacial water. However, it can be seen that the interfacial water orientation in hydrophilic [BMIM][BF₄] differs a lot from that in two hydrophobic RTILs, suggesting a dominant influence of anion type on water adsorption. In addition, water orientations under positive polarization with respect to three different RTILs are very similar.

Figure R2.11 | Influence of ion type on interfacial water at gold electrodes. **a-c**, Orientation of the interfacial water for RTILs [pyr14][TFSI] (**a**), [BMIM][TFSI] (**b**), and [BMIM][BF₄] (**c**) under an applied potential of 4 V. **d-e**, Hydrogen bonds formed by RTIL cations (**d**) and anions (**e**) with interfacial water as a function of applied potential. **f**, The cartoon of angles showing water orientation: water dipole orientation, θ_{dipole} , is defined as the angle formed by the water vector and the normal of electrode surface; and water normal orientation, θ_{normal} , is defined as the angle formed between the normal of water plane and the normal of electrode surface. Red solid lines are for water dipole orientation, and blue dashed lines represent results for water normal orientation. The definition of H-bond can be found in the **Supplementary Information**.

Specifically, near negatively charged gold electrode, the dipole orientation of interfacial water molecules in the two hydrophobic RTILs (peak located at 152° and 135° for [pyr14][TFSI] and [BMIM][TFSI], respectively) as well as their normal orientation (peak located at 90°) show that most water tends to be vertical to the electrode with its two hydrogen atoms both approaching to the surface (see **Figure R2.11a** and **R2.11b**). The counterpart figure for water in hydrophilic [BMIM][BF₄] illustrates that water is also vertical to the electrode (normal orientation features at 90°), however, it is more tilted than water in [pyr14][TFSI]/[BMIM][TFSI] as one of its hydrogen atom is closer to the surface than the other (dipole orientation shows two probability peak located at 90° and 154° , respectively). Near positively charged electrode, for all three RTILs, their similar normal orientations showing two peaks located at 16° and 164° as well as the dipole orientations with peak location at about 75° depict that most water adopts a configuration parallel to the electrode.

Changing electrode type from gold to carbon does not alter much the trends in these observations (see **Supplementary Fig. 12**), which further suggests a secondary role of electrode in water electrosorption. Thus, how does the association/interaction between water and cations/anions affect the structure of interfacial water? To answer the reviewer's question, the number of hydrogen bonds with respect to the EDL potential for three humid RTILs have been calculated and compared in **Figure R2.11d-e**. It can be seen that: 1) under potential of zero charge, the number of hydrogen bonds formed between water and anion is much larger than that of water and cation, regardless of the type of RTIL. 2) BMIM⁺ ions and BF₄⁻ ions have stronger hydrogen bonding with water than the pyr14⁺ and TFSI⁻ over the whole EDL potential range, respectively. These observations are in line with previous studies pointing that water molecules do associate with cation; however, they have stronger interaction with anion, which also account for why anions mainly determine the miscibility of RTILs.^{11,12}

When the electrode surface becomes more negatively electrified, for hydrophobic [pyr14][TFSI] the number of water–pyr14⁺ H-bonds remains relatively small and stable, and that of water–TFSI⁻ H-bonds decreases dramatically to a small value (e.g., from 1.12 to 0.10 on gold electrode). The scenario of [BMIM][BF₄] is much different, since water and BF₄⁻ still forms more hydrogen bonds than water–BMIM⁺ even in a cation-rich double layer (for instance, 0.8 vs. 0.46 on gold electrode, at an EDL potential of $\sim -2V$). These findings could help to understand why hydrophobic [pyr14][TFSI] and hydrophilic [BMIM][BF₄] could result in very different

water orientations near negative polarization, that is, 1) in the interfacial region under negative polarization, hydrogen bonding of water–[BMIM][BF₄] is much stronger than water–[pyr14][TFSI]; 2) Although BF₄[−] ions are squeezed away from the negatively charged electrode, they are still strongly associated with interfacial water molecules in terms of hydrogen bonds so that hydrogen atoms of water are attracted to point to the fluorine atoms.

When EDL potential gets more positive, the number of water–anion H-bonds increases a little bit, while each water could form fewer hydrogen bonds with cations (for [BMIM][BF₄]) or maintain at a quite low level (for [pyr14][TFSI]). This can be ascribed to the anion accumulation and cation removal in the double layer near positively charged electrodes. From normal orientation of the interfacial water under positive polarization shown in the **Figure R2.11a-c**, water tends to lie on the electrode surface, as water forms strong hydrogen bonding with the surrounding anions.

In conclusion, MD-obtained orientations of interfacial water and H-bonds based analysis suggest that the strong association between water–BF₄[−] exists even under negative polarization (in which anions are co-ions), resulting in unstable water electroadsorption on negatively charged electrode. Water interacts more strongly with BMIM⁺ than pyr14⁺; however, the effects of cation type are minor, as the influence of cation type on the orientation of interfacial water is weaker compared with anion type effects.

Brief summary:

Discussions on 1) the orientation of interfacial water and 2) the hydrogen bonding between interfacial water and nearby, have been integrated into the revised manuscript (*Results Section of main text, pages 11-12, lines 259-283; page 16, lines 376-382*)

“As PMFs demonstrate where water would be predominantly distributed, we then focus on the structures of water within the interfacial region. Since water in hydrophobic and hydrophilic RTILs undergoes different association with cations/anions, one can expect that the structures of interfacial water may depend on the nature of the RTILs, EDL potential and electrode type. Revealed by the orientations of water in [pyr14][TFSI] and [BMIM][BF₄] at the electrified electrode (Supplementary Fig. 12), it could be concluded that 1) the interfacial water tends to orient vertically to the negative electrode but lies flat on the positive electrode; 2) for negatively charged electrode water orientation in different RTILs exhibits some differences, but no notable

differences are seen at the positively charged electrode. Since the surface charges would always force the dipole of water to align vertically to electrode surface, the discrepancy of water orientation at cathode and anode would be attributed to the different water-RTIL associations, to a large extent, determined by water-ion hydrogen bonding. Detailed analysis of hydrogen bonding between water and cations/anions in the interfacial region, which can be found in Part G of Supplementary Information (Supplementary Fig. 13), unveils that 1) water forms more H-bonds with anions than cations, even in the cation-rich EDL under negative polarization, suggesting that anions play a dominant role in hydrophobicity/hydrophilicity of RTILs; 2) more water-ion H-bonds are observed in hydrophilic [BMIM][BF₄] than in hydrophobic [pyr14][TFSI], consistent with previous work.^{45,46}

The overall water-RTIL hydrogen bonding as well as water orientation strongly depends on the potential drop across the EDL. Thus, naturally, the electrosorption of water depends not only on the water-ion interaction, ‘water following anions’, but also on the water-electrode interaction.”

“The comparison of three RTILs studied – [pyr14][TFSI], [BMIM][TFSI], and [BMIM][BF₄], in terms of the interfacial water orientation and hydrogen bonding with RTIL ions reveals that it is anion type (which dominates the hydrophilicity of RTIL) rather than cation type that majorly determines the enrichment or depletion of water electrosorption at the negative electrode (Supplementary Fig. 16). Indeed, more RTILs should be tested to prove the universality of this effect, but its physical nature seems to be very sound.”

New references are added, as:

45. Cammarata, L., Kazarian, S. G., Salter, P. A. & Welton, T. Molecular states of water in room temperature ionic liquids. *Phys. Chem. Chem. Phys.* **3**, 5192–5200 (2001).

46. Anthony, J. L., Maginn, E. J. & Brennecke, J. F. Solution thermodynamics of imidazolium-based ionic liquids and water. *J. Phys. Chem. B* **105**, 10942–10949 (2001).”

More detailed descriptions related to this comment have been addressed in the updated **Supplementary Information (Supplementary Part G. Orientation and H-bond of water adsorbed at the electrode; Part H. Explorations on additional RTILs)**.

Comment 7) Figure 3b: The overall current level is about 10x less compared to the 3a. This should be emphasized. My question here is: Why is the current on the anodic side so much lower as well. The argument of the authors is, that the hydrophilic IL suppresses the cathodic water

adsorption but not the anodic side (at least in the simulations). The anodic side seems similar in simulations. Hence I would expect a similar current and significant gold oxidation/reduction in experiments as well. This is not observed. Why?

Response:

Thank the reviewer for the careful reading and the comment.

The difference in overall current level for two different RTILs (in Figure 3 of previous version of manuscript) resulted from the abrupt change of current around the upper and lower limits of the potentials, that is, slightly changed potential limit can result in large change of the current. When we conducted new experiments for obtaining data to update Figure 3, the lower and upper limits of electrochemical windows were better controlled. Therefore, the current level for **Figure 3a** is comparable with that of **Figure 3b** as shown in the updated **Figure 3** in the revised main text.

For the reviewer's question that why the current on the anodic side is lower or not similar, we agree with the reviewer that the hydrophilic IL does not suppress the anodic water adsorption, but unlike water reduction, water oxidation needs much larger overpotential,²¹ which leads to the missing of water oxidation on CVs. Surely, as expected by the reviewer, gold oxidation/reduction indeed appears on CVs at anodic side. It is reasonable that the potentials and wave shapes of gold oxidation/reduction are different in two ILs with or without water, especially when the oxidation/reduction of ILs is involved (see **Figure R2.7** in **Comment 3**). For instance, in the case of humid [pyr14][TFSI] with 1260 ppm water content, an anodic current wave at around +2.0 V and a cathodic peak at about +1.4 V were observed, which stand for the oxidation of Au(111) surface and the corresponding reduction, respectively.

Brief summary:

In response to the reviewer's instructive comment, we have revised the manuscript (*Results Section of main text, page 8-9, lines 200-204*), as:

“When potential scans positively, the characteristic peaks are present. Taking 1260 ppm water content as an example, an anodic current wave at around +2.0 V and a cathodic peak at about +1.4 V were observed, which stand for the oxidation of Au(111) surface and the corresponding reduction, respectively.”

Minor aspects:

Comment 1) In figure 1, the potential axis is a bit misleading, 1b and 1d go from negative to positive, while 1a and 1c go from positive to negative. Please emphasize the inversion of the axis orientation in the figure chart and/or caption.

Response:

Thank the reviewer for the helpful suggestion. We change the figures ALL with the potential axis from the positive to negative, which would render a better display than previous, in particular for readers directly making comparisons among ions.

Brief summary:

Figure 1 and the **Supplementary Figure 4** are updated based on the reviewer's advice, as:

Figure R2.11 | Ion distribution in RTIL near electrodes with varying applied potential: the effect of the electrode material. Ions number density as a function of distance from the electrode under different EDL potential. **a-d**, pyr14⁺ (**a**) and TFSI⁻ (**b**) number density profiles near gold surface. pyr14⁺ (**c**) and TFSI⁻ (**d**) number density profiles near carbon surface. Blue and red in floor colors highlight negative and positive potential bias, respectively. Note that planes across centers of surface atoms of carbon and gold electrodes are located at Z = 0 nm, but the results are displayed in the relevant range of 0.2 nm < Z < 1 nm. Through this work, number density is computed based on the center of mass of each ion.

And

Figure R2.12 | Ion number density as a function of distance from the electrode with varying the EDL potential. a-d, BMIM⁺ (a) number density and BF₄⁻ (b) number density on gold electrodes. BMIM⁺ (c) number density and BF₄⁻ (d) number density on carbon electrodes. Blue and red in floor color correspond to negative and positive potential respectively. Carbon and gold electrodes are located at Z = 0 nm, however, Z axis limits are specified from 0.2 nm to 1 nm for better illustration.

Comment 2) According to the experiment procedure, all experiments were conducted in an Ar-filled glovebox. Humid ionic liquids are prepared with mixing dry ionic liquid with water. Did the authors measure the water evaporation rate from the ionic liquid system into the glovebox? In previous work by other authors, the amount of water in ionic liquid systems is usually equilibrated with the surrounding humidity, which allows to generate stable values. Here, water levels in the IL may drop/ change with time. Such effects should be considered and ideally also measured/ estimated.

Response:

We thank the reviewer for his/her helpful comment.

Some measures are usually taken to remove the absorbed water from ILs, for example, ILs can be vacuum-dried overnight at 80 degrees Celsius in a glovebox. Then water concentration can be less than 50 ppm, but is still larger than water concentration in glovebox (less than 1 ppm). However, on exposure to ambient humidity, ILs can quickly absorb moisture. We measured the change of water concentration in the two ILs with the time, as shown in **Figure R2.1**. It can be seen that water concentrations in both ILs increase with the time of expose to air. In about two days, water concentration reaches to ~3500 and 30000 ppm for [pyr14][TFSI] and [Bmim][BF₄],

respectively. After exposure to air for one week, water concentrations are ~3900 and ~33000 ppm for [pyr14][TFSI] and [Bmim][BF₄], respectively.

The new investigations on the impact from water concentration, by both MD simulation and experiment, have showed that the influence of water concentration on the observations in **Figures 2-3** in the main text is weak. Therefore, even there is a deviation in measuring water content, the key conclusion would still hold as: water molecules in hydrophobic RTILs ([pyr14][TFSI]) eagerly adsorb onto the electrified electrode surface and thus favor the electrochemical reaction of water, while water molecules in hydrophilic RTILs ([BMIM][BF₄]) prefer keeping a distance away from the negatively polarized electrode surface, which could hinder the electrochemical reduction of water.

Nevertheless, we indeed agree with the reviewer that water concentration during the experiments should be kept stable. Indeed, the electrochemical cell we used for measurements was a sealed one, which was isolated from the environment, and CV measurement was usually completed within half an hour. The above can keep water level stable to a great extent.

Brief summary:

In response to the reviewer's comment, we have revised the manuscript (*Method Section of main text, page 20, lines 490-496*), as:

“The electrochemical cell used for measurements was a sealed one, which was isolated from the environment, and CV measurement was typically completed within half an hour, so that the water concentration level was kept stable.”

Responses to Reviewer #3's Comments

Reviewer #3 (Remarks to the Author):

This is an interesting MD study (coupled with some experimental work) on an important problem of water adsorption/desorption on electrified surfaces in the presence of ionic liquid electrolytes. The authors found that the hydrophilic ionic liquid [BMIM][BF₄] leads to water depletion at the negative electrode as the magnitude of its electrode potential increases while the hydrophobic ionic liquid [pyr14][TFSI] results in the enhancement of water density at the electrode surface irrespective of its polarization. This is in agreement with their experimental observation that water is not reduced in [BMIM][BF₄] at the negative electrode. I have following comments that I suggest the authors should consider before the manuscript becomes acceptable for publication:

Response:

We thank the reviewer for carefully reading and positively assessing our work and for providing us with the opportunity to amend our manuscript.

Comment (1) I suggest that error bars be calculated and displayed in Figures 2 and 5. It is known that the convergence of ionic liquid simulations is very slow, but the authors appear to have decent MD statistics. Therefore the presentation of error bars will strengthen the quality of their analysis and give the reader good perspective on the authors' MD results.

Response:

Thanks the reviewer for this helpful comment. We have added error bar for **Figure 2** and **Figure 5** in the main text. The error bar for each point is the standard deviation of the results for three independent simulation cases.

Brief summary:

Figure 2 and **Figure 5** in the main text have been updated with adding error bars, as:

Figure R3.1 | Electrosorption of water from humid RTILs on gold and carbon electrodes. Panels (a) and (b) show water adsorption from humid [pyr14][TFSI] and [BMIM][BF₄], respectively; results for gold electrode are shown by blue triangles and for carbon electrode by red circles. The relative enrichment or depletion of water molecules in the first interfacial layer is depicted by the ratio of averaged number density of water in this layer to that of water in the bulk region. Enrichment (depletion) zone corresponds to a higher (lower) water density than in the bulk. **The error bar for each point is the standard deviation of the results based on three independent simulation cases.**

And

Figure R3.2 | The weight of different components of the total interaction energy of water molecule adsorbed at the electrode surface under different electrode polarizations. a-d, The figures for water adsorbed from [pyr14][TFSI] (a) and [BMIM][BF₄] (b) near gold electrode and from [pyr14][TFSI] (c) and [BMIM][BF₄] (d) near carbon electrode. Red, yellow, blue, and green bar charts represent water-ions van der Waals, water-ions coulombic, water-electrode van der Waals, and water-electrode coulombic interactions, respectively. The magenta dots connected by solid lines are the total value of “local” potential energy of water with its nearby species in the interfacial region, which results from the summation of previous four interactions. **The error bar for each magenta dot is the standard deviation of the results for three independent simulation cases.**

Comment (2) I wonder how the electronic polarizability of water and ionic liquids will influence the results, in particular, the interaction energies in Figure 5. I could be wrong but I suspect that water at the electrode surface would be affected most. Furthermore, its effect would probably increase with the magnitude of the electrode potential. Some comments are warranted on this issue.

Response:

We thank the reviewer for this quite helpful comment.

In general, the electronic polarizability of water and RTIL molecules should be accounted for accurately characterizing their mutual interactions and dynamics. However, previous simulations based on non-polarizable water/RTIL models have successfully captured key features of the water/RTIL mixture.²²⁻²⁴ Therefore, the classical force fields adopted here are deemed adequate. In addition, the electronic polarizability of the electrode and the image charge effects are taken into account in these constant-potential simulations, so that one can describe by the most accurate coulombic interaction between electrode and water or ions.

To the best of our knowledge, explicit account for electronic polarizability of water is crucial in describing kinetic properties such as proton transport or self-diffusion of water molecules.^{25,26} However, it is known to be less important for equilibrium properties that can be emulated with effective force fields that indirectly take into account the electronic polarizability of water and ions.²⁷ Again testing this approximation would be the best done with future *ab initio* calculations (force-field free simulations).

Brief summary:

In response to this comment, we have added two instructive content on this to the revised manuscript (*Method Section of main text, page 18, lines 435-440; page 19, lines 450-454*), as:

“In theory, the electronic polarizability of water molecules and RTIL ions should be accounted for accurately characterizing their mutual interactions and dynamics. However, previous simulations based on non-polarizable water/RTIL models have successfully captured key features of the water-RTIL mixture.^{46,56,57} Therefore, the classical force fields adopted here are deemed adequate.”

“It is worthy to note that CPM by far can mainly focus on the electrolyte side of the interface and cannot adequately describe the potential distribution in the electrode (electrode is modeled as a conductor). Therefore, the levels of ‘metallicity’ of gold and carbon electrodes are not taken into account in this work.”

New references to previous works on simulating water/RTIL mixture are added, as:

45. Cammarata, L., Kazarian, S. G., Salter, P. A. & Welton, T. Molecular states of water in room temperature ionic liquids. *Phys. Chem. Chem. Phys.* **3**, 5192–5200 (2001).

56. Jiang, W., Wang, Y. & Voth, G. A. Molecular dynamics simulation of nanostructural organization in ionic liquid/water mixtures. *J. Phys. Chem. B* **111**, 4812–4818 (2007).

57. Hayes, R., Imberti, S., Warr, G. G. & Atkin, R. How water dissolves in protic ionic liquids. *Angew. Chemie - Int. Ed.* **51**, 7468–7471 (2012).”

References

1. Gerischer, H. An interpretation of the double layer capacity of graphite electrodes in relation to the density of states at the Fermi level. *J. Phys. Chem.* **89**, 4249–4251 (1985).
2. Randin, J. & Yeager, E. Differential Capacitance Study of Stress-Annealed Pyrolytic Graphite Electrodes. *J. Electrochem. Soc.* **118**, 711–714 (1971).
3. Hahn, M. *et al.* Interfacial Capacitance and Electronic Conductance of Activated Carbon Double-Layer Electrodes. *Electrochem. Solid-State Lett.* **7**, A33–A36 (2004).
4. Kornyshev, A. A., Luque, N. B. & Schmickler, W. Differential capacitance of ionic liquid interface with graphite: the story of two double layers. *J. Solid State Electrochem.* **18**, 1345–1349 (2014).
5. Fang, T., Konar, A., Xing, H. & Jena, D. Carrier statistics and quantum capacitance of graphene sheets and ribbons. *Appl. Phys. Lett.* **91**, 40–43 (2007).
6. Xu, H., Zhang, Z. & Peng, L. M. Measurements and microscopic model of quantum capacitance in graphene. *Appl. Phys. Lett.* **98**, 2009–2012 (2011).
7. Xia, J., Chen, F., Li, J. & Tao, N. Measurement of the quantum capacitance of graphene. *Nat. Nanotechnol.* **4**, 505–509 (2009).
8. Lee, A. A., Vella, D. & Goriely, A. Quantum capacitance modifies interionic interactions in semiconducting nanopores. *EPL* **113**, 38005 (2016).
9. Car, R. & Parrinello, M. Unified Approach for Molecular Dynamics and Density-Functional Theory. *Phys. Rev. Lett.* **55**, 2471–2474 (1985).
10. Vlcek, L. *et al.* Electric Double Layer at Metal Oxide Surfaces: Static Properties of the Cassiterite–Water Interface. *Langmuir* **23**, 4925–4937 (2007).
11. D’Angelo, P., Zitolo, A., Aquilanti, G. & Migliorati, V. Using a Combined Theoretical and Experimental Approach to Understand the Structure and Dynamics of Imidazolium-Based Ionic Liquids/Water Mixtures. 2. EXAFS Spectroscopy. *J. Phys. Chem. B* **117**, 12516–12524 (2013).
12. Porter, A. R., Liem, S. Y. & Popelier, P. L. A. Room temperature ionic liquids containing low water concentrations—a molecular dynamics study. *Phys. Chem. Chem. Phys.* **10**, 4240–4248 (2008).
13. Cao, Y., Chen, Y., Sun, X., Zhang, Z. & Mu, T. Water sorption in ionic liquids: kinetics, mechanisms and hydrophilicity. *Phys. Chem. Chem. Phys.* **14**, 12252–12262 (2012).
14. Baj, S., Chrobok, A. & Słupska, R. The Baeyer–Villiger oxidation of ketones with bis(trimethylsilyl) peroxide in the presence of ionic liquids as the solvent and catalyst. *Green*

- Chem.* **11**, 279–282 (2009).
15. Malik, R. S., Tripathi, S. N., Gupta, D. & Choudhary, V. Novel anhydrous composite membranes based on sulfonated poly (ether ketone) and aprotic ionic liquids for high temperature polymer electrolyte membranes for fuel cell applications. *Int. J. Hydrogen Energy* **39**, 12826–12834 (2014).
 16. Huddleston, J. G. *et al.* Characterization and comparison of hydrophilic and hydrophobic room temperature ionic liquids incorporating the imidazolium cation. *Green Chem.* **3**, 156–164 (2001).
 17. Rivera-Rubero, S. & Baldelli, S. Influence of water on the surface of hydrophilic and hydrophobic room-temperature ionic liquids. *J. Am. Chem. Soc.* **126**, 11788–11789 (2004).
 18. Gutowski, K. E. *et al.* Controlling the Aqueous Miscibility of Ionic Liquids: Aqueous Biphasic Systems of Water-Miscible Ionic Liquids and Water-Structuring Salts for Recycle, Metathesis, and Separations. *J. Am. Chem. Soc.* **125**, 6632–6633 (2003).
 19. Galiński, M., Lewandowski, A. & Stepniak, I. Ionic liquids as electrolytes. *Electrochim. Acta* **51**, 5567–5580 (2006).
 20. Li, Q. *et al.* The electrochemical stability of ionic liquids and deep eutectic solvents. *Sci. China Chem.* **59**, 571–577 (2016).
 21. McCrory, C. C. L. *et al.* Benchmarking Hydrogen Evolving Reaction and Oxygen Evolving Reaction Electrocatalysts for Solar Water Splitting Devices. *J. Am. Chem. Soc.* **137**, 4347–4357 (2015).
 22. Anthony, J. L., Maginn, E. J. & Brennecke, J. F. Solution thermodynamics of imidazolium-based ionic liquids and water. *J. Phys. Chem. B* **105**, 10942–10949 (2001).
 23. Jiang, W., Wang, Y. & Voth, G. A. Molecular dynamics simulation of nanostructural organization in ionic liquid/water mixtures. *J. Phys. Chem. B* **111**, 4812–4818 (2007).
 24. Hayes, R., Imberti, S., Warr, G. G. & Atkin, R. How water dissolves in protic ionic liquids. *Angew. Chemie - Int. Ed.* **51**, 7468–7471 (2012).
 25. Walbran, S. & Kornyshev, A. A. Proton transport in polarizable water. *J. Chem. Phys.* **114**, 10039–10048 (2001).
 26. Kornyshev, A. A., Kuznetsov, A. M., Spohr, E. & Ulstrup, J. Kinetics of proton transport in water. *J. Phys. Chem. B* **107**, 3351–3366 (2003).
 27. Leontyev, I. & Stuchebrukhov, A. Accounting for electronic polarization in non-polarizable force fields. *Phys. Chem. Chem. Phys.* **13**, 2613–2626 (2011).

Reviewers' Comments:

Reviewer #1:

Remarks to the Author:

The authors have well addressed my various comments. I now recommend the publication of this manuscript in Nature Communications.

Reviewer #2:

Remarks to the Author:

The authors have done a good job in revising this manuscript according to the suggestions and comments of all three referees. Specifically, the authors now also include a number of additional control experiments and they also remeasured and further analyzed some of their previous data. The new data along with the more detailed analysis of the MD simulations now supports the generalizing conclusions of the authors, and I feel that this work is now ready for publication in Nature Communications.

Reviewer #3:

Remarks to the Author:

The authors have addressed my concerns/questions reasonably in the revised version of the manuscript. I thus recommend publication of the manuscript.

We've made point-to-point response to each comment. Specifically, reviewers' comments are copied in blue, and each comment is followed by our response in black.

REVIEWERS' COMMENTS:

Reviewer #1 (Remarks to the Author):

The authors have well addressed my various comments. I now recommend the publication of this manuscript in Nature Communications.

Response: We thank this reviewer for her/his positive comment and recommendation to Nature Communications.

Reviewer #2 (Remarks to the Author):

The authors have done a good job in revising this manuscript according to the suggestions and comments of all three referees. Specifically, the authors now also include a number of additional control experiments and they also remeasured and further analyzed some of their previous data. The new data along with the more detailed analysis of the MD simulations now supports the generalizing conclusions of the authors, and I feel that this work is now ready for publication in Nature Communications.

Response: We appreciate that the reviewer re-evaluated our work, and recommended our revised manuscript for publication in Nature Communications.

Reviewer #3 (Remarks to the Author):

The authors have addressed my concerns/questions reasonably in the revised version of the manuscript. I thus recommend publication of the manuscript.

Response: We appreciate the reviewer for her/his time and work on re-reviewing our work and the recommendation for publication.